# EXP1 is critical for nutrient uptake across the parasitophorous vacuole membrane of malaria parasites

Paolo Mesén-Ramírez[1], Bärbel Bergmann[1], Thuy Tuyen Tran[1], Matthias Garten[2], Jan Stäcker[1], Isabel Naranjo-Prado[1], Katharina Höhn[3], Joshua Zimmerberg[2], Tobias Spielmann[1‡*]

**1** Molecular Biology and Immunology Section, Bernhard Nocht Institute for Tropical Medicine, Hamburg, Germany, **2** Section on Integrative Biophysics, Eunice Kennedy Shriver National Institute of Child Health and Human Development, National Institutes of Health, Bethesda, Maryland, United States of America, **3** Electron Microscopy Unit, Bernhard Nocht Institute for Tropical Medicine, Hamburg, Germany

‡ TS is the lead author for this work.
* spielmann@bni-hamburg.de

**Data Availability Statement:** All relevant data are within the paper and its Supporting Information files.

## Abstract

Intracellular malaria parasites grow in a vacuole delimited by the parasitophorous vacuolar membrane (PVM). This membrane fulfils critical roles for survival of the parasite in its intracellular niche such as in protein export and nutrient acquisition. Using a conditional knockout (KO), we here demonstrate that the abundant integral PVM protein exported protein 1 (EXP1) is essential for parasite survival but that this is independent of its previously postulated function as a glutathione S-transferase (GST). Patch-clamp experiments indicated that EXP1 is critical for the nutrient-permeable channel activity at the PVM. Loss of EXP1 abolished the correct localisation of EXP2, a pore-forming protein required for the nutrient-permeable channel activity and protein export at the PVM. Unexpectedly, loss of EXP1 affected only the nutrient-permeable channel activity of the PVM but not protein export. Parasites with low levels of EXP1 became hypersensitive to low nutrient conditions, indicating that EXP1 indeed is needed for nutrient uptake and experimentally confirming the long-standing hypothesis that the channel activity measured at the PVM is required for parasite nutrient acquisition. Hence, EXP1 is specifically required for the functional expression of EXP2 as the nutrient-permeable channel and is critical for the metabolite supply of malaria parasites.

## Introduction

Malaria pathology results from the replication of *Plasmodium* parasites in red blood cells (RBCs). Malaria parasites proliferate within a parasitophorous vacuole (PV) surrounded by the membrane of the PV (PVM) [1,2], the interface between parasite and host cell. The PVM harbors protein complexes that serve key functions for the intracellular survival of the parasite [1]. The *Plasmodium* translocon of exported proteins (PTEX) mediates the transport of

**Funding:** This work was supported by the German Research Foundation (grant SP1209/4-1 to TS) (https://www.dfg.de/en/) and the Division of Intramural Research of the Eunice Kennedy Shriver National Institute of Child Health and Human Development, National Institutes of Health (https://www.nichd.nih.gov/) to JZ. JS is supported by a fellowship of the Jürgen Manchot Stiftung (https://www.im.nrw/juergen-manchot-stiftung). INP was supported by a scholarship from Colciencias (https://www.colciencias.gov.co/). The funders had no role in study design, data collection and analysis, decision to publish, or preparation of the manuscript.

**Competing interests:** The authors have declared that no competing interests exist.

**Abbreviations:** 5-ALA, 5-aminolevulinic acid; ART, artemisinin; BIP, binding immunoglobulin protein; BSA, bovine serum albumin; CM2-DCFDA, chloromethyl-dichlorofluorescein; DIC, differential interference contrast; DHA, dihydroartemisinin; DiCre, Dimerizable Cre; ETRAMP, early transcribed membrane protein; EXP1, exported protein 1; FC, flow cytometry; GSH, reduced glutathione; GST, glutathione S-transferase; h.p.i., hours post invasion; HSP101, heat shock protein 101; IC50, half maximal inhibitory concentration; IFA, immunofluorescence assay; IP, immunoprecipitation; KAHRP, knob-associated histidine-rich protein; KO, knockout; GFP, green fluorescent protein; MAPEG, membrane-associated proteins in eicosanoid and glutathione metabolism; MSRP6, MSP7-related protein 6; MSP1, merozoite surface protein 1; PAGE, polyacrylamide gel electrophoresis; PbEXP1, *P. berghei* EXP1; PfEXP1, *P. falciparum* EXP1; PPIX, protoporphyrin IX; PPM, parasite plasma membrane; PSAC, parasite surface anion channel; PTEX, *Plasmodium* translocon of exported proteins; PV, parasitophorous vacuole; PVM, parasitophorous vacuolar membrane; RBC, red blood cell; REX1, ring exported protein 1; ROS, reactive oxygen species; RPMI, Roswell Park Memorial Institute; RSA, ring-stage survival assay; SBP1, skeleton binding protein 1; SDS, sodium dodecyl sulfate; SERA5, serine-rich antigen 5; SLI, selection-linked integration; TM, transmembrane.

parasite proteins across the PVM into the host cell [3,4,5]. A nonselective pore permits passage of nutrients such as monosaccharides, folates, and amino acids through the PVM [6,7]. While the composition and function of PTEX has been studied in some detail, the molecular basis for the nutrient-permeable channel activity is much less defined. PTEX comprises oligomers of 3 core components, including exported protein 2 (EXP2), which forms a heptameric PVM-spanning channel [8]. Conditional knockdown and patch-clamp measurements indicated that EXP2 has a dual role in both protein export as part of PTEX and in the nutrient-permeable channel activity of the PVM [7], a conclusion initially indicated by the homology and complementation capacity of EXP2 to 2 proteins required for the solute pore activity in *Toxoplasma gondii* [9].

Other PVM proteins include highly expressed single-pass transmembrane (TM) proteins such as early transcribed membrane proteins (ETRAMPs) [10] and exported protein 1 (EXP1) [11]. EXP1 was the first protein localized to the PVM of blood and liver stages where it forms homo-oligomers with the N-terminus facing the PV lumen and the C-terminus exposed to the RBC cytosol [1,11,12,13]. Unsuccessful attempts to genetically disrupt *exp1* suggested an essential role for EXP1 in parasite development [14].

Bioinformatic analyses classified EXP1 as a member of the superfamily of membrane-associated proteins in eicosanoid and glutathione metabolism (MAPEG) [15,16]. Enzymatic in vitro assays indicated that EXP1 possesses glutathione S-transferase (GST) activity, and it was proposed to protect the parasite from oxidative damage via detoxification of hemoglobin byproducts by conjugation with reduced glutathione (GSH) [16]. It was further proposed that EXP1 is associated with artemisinin (ART) resistance, as its transcription was up-regulated in ART-resistant parasite strains [16]. A study in *P. berghei* indicated that EXP1 may also play a role in the uptake of lipids during intrahepatic development [17].

Here, using a conditional knockout (KO) of *exp1*, we show that EXP1 is essential for the growth of *P. falciparum* blood stages independently of the previously proposed GST activity. We find that EXP1 is required for the nutrient-permeable channel activity of the PVM. Consistently, parasites with reduced levels of EXP1 became hypersensitive to nutrient-limiting conditions, associating the permeability measured at the PVM with nutrient acquisition. EXP1 interacts with EXP2 at the PVM and is required for EXP2's proper distribution and function as a nutrient-permeable channel but not for the function of EXP2 in protein export. Hence, EXP1 defines the function of EXP2 as a nutrient-permeable channel and is critical for nutrient uptake across the PVM, which now can be specifically studied.

## Results

### EXP1 is essential for parasite development in RBCs

To first test whether EXP1 is required for the survival of *P. falciparum* erythrocytic stages, we generated a conditional *exp1* KO based on the Dimerizable Cre (DiCre) system [18,19, 20] using selection-linked integration (SLI) [20]. The endogenous *exp1* was disrupted before the region encoding the TM domain, and at the same time, a second functional copy of *exp1* flanked by loxP sites was introduced in the *exp1* locus. The new floxed copy of EXP1 is hemagglutinin (HA) tagged and can be conditionally excised by DiCre upon addition of rapalog (Fig 1A and S1A and S1B Fig). Immunofluorescence assays (IFAs) showed that the corresponding cell line (condΔEXP1) correctly expressed the functional EXP1-HA in the PVM (Fig 1B).

To investigate the effect of the loss of EXP1 on parasite survival, synchronous condΔEXP1 ring-stage parasites were grown in the presence of rapalog, alongside a control culture (Fig 1C). PCR confirmed efficient excision of the functional copy of *exp1* within one growth cycle (48 hours) upon addition of rapalog (Fig 1C). No growth defect was observed in this first cycle

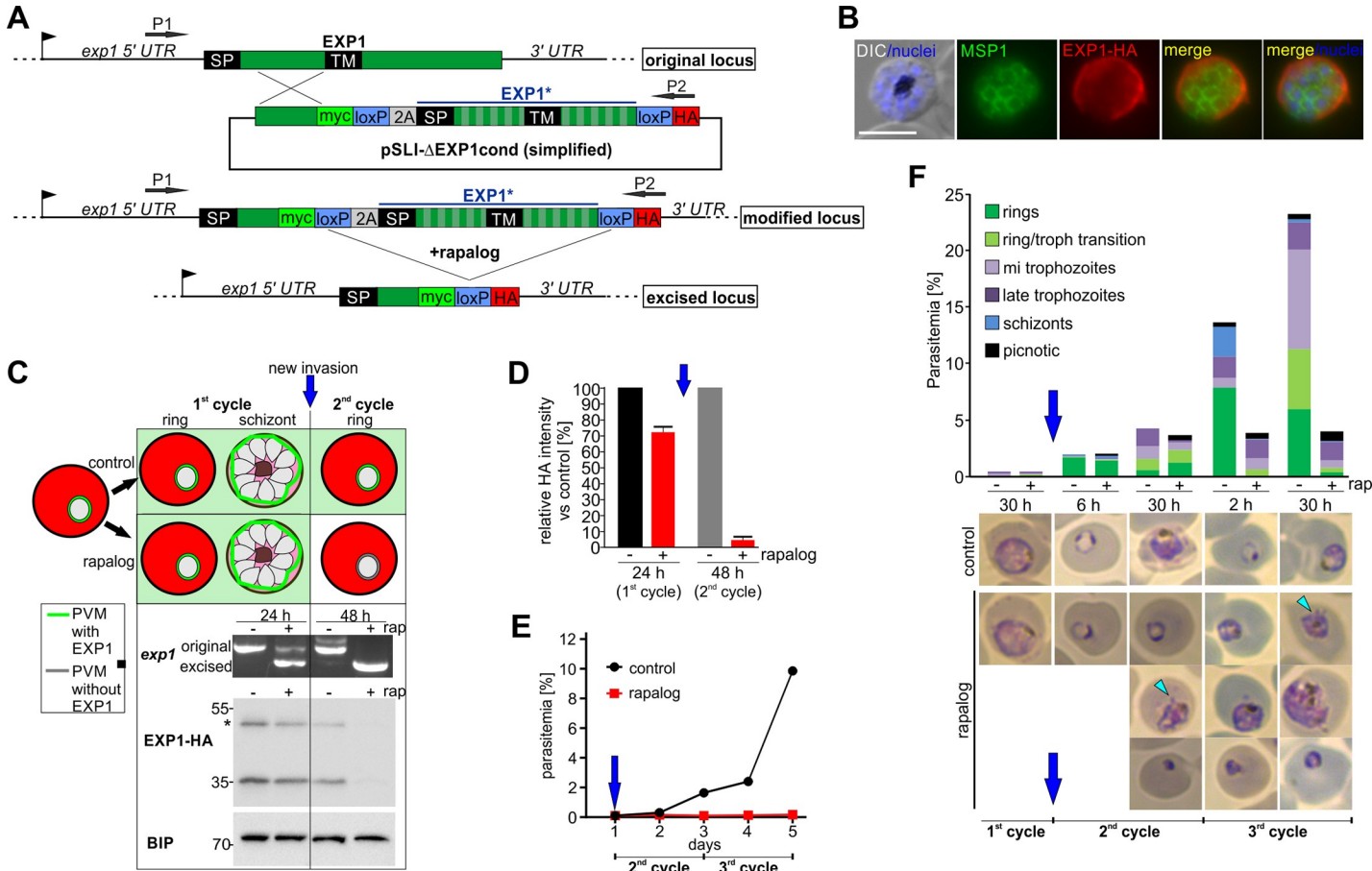

**Fig 1. EXP1 is essential for blood stage development.** (A) Simplified schematic of DiCre-based conditional *exp1* KO using SLI. Arrows indicate primers P1 and P2 (see S1 Fig for details). (B) IFA of compound 2-arrested condΔEXP1 late-stage schizonts show localization of EXP1*-HA at the PVM. α-MSP1 (MSP1) labels the PPM. Nuclei were stained with DAPI; scale bars, 5 μm. (C) Strategy for depletion of EXP1 from the PVM of synchronized condΔEXP1 ring stages divided into a culture with rapalog ("rap") and one without rapalog ("control"). Top: schematic: green boxes and green line signify PVM with EXP1. Mid: PCR with primers P1 and P2 from gDNA 24 and 48 hours after addition of rapalog. Original: band for intact *exp1* locus (1,919 bp); excised: band after excision of *exp1** (1,326 bp). Bottom: western blot probed with α-HA to detect EXP1*-HA and α-BIP as loading control. Asterisk: unskipped (first T2A product that is present before excision and hence has no impact on parasites. (D) Quantification of levels of wtEXP1-HA in the first ("24 h") and second ("48 h") cycle after addition of rapalog based on densitometric analysis of anti-HA immunoblots (mean of *n* = 2 independent experiments) of which one is shown in (C). HA signal was normalized to BIP. Error bars indicate SD. (E) FC growth curves of synchronous ring stage condΔEXP1 parasites starting in cycle 2 (ΔEXP1 parasites) as shown in (C). One representative of *n* = 5 independent experiments. (F) Stage distribution of condΔEXP1 parasites in Giemsa smears of synchronous parasites grown with (+) and one without (−) rapalog ("rap") at different time points (average time post invasion) after adding rapalog. Light blue arrowheads show blebs. One of *n* = 4 independent experiments is shown. Blue arrow in (C–F) indicates start of a new cycle without EXP1. 2A, T2A skip peptide; BIP, binding immunoglobulin protein; DIC, differential interference contrast; DiCre, Dimerizable Cre; EXP1, exported protein 1; EXP1*, recodonized *exp1*; FC, flow cytometry; gDNA, genomic DNA; HA, triple hemagglutinin tag; KO, knockout; MSP1, merozoite surface protein 1; PPM, parasite plasma membrane; PVM, parasitophorous vacuolar membrane; SLI, selection-linked integration; SP, signal peptide; TM, transmembrane domain; wt, wild-type.

during which the *exp1* gene was excised (S1C Fig). In this cycle, EXP1 protein levels remained at approximately 70% due to protein expressed in the PVM before excision was complete (Fig 1C and 1D and S1D Fig). Western blot confirmed loss of EXP1 (approximately 5% residual protein detected at 48 h; Fig 1D) in parasites (henceforth termed ΔEXP1 parasites) after invasion and start of a new cycle (Fig 1C, blue arrow). The ΔEXP1 parasites failed to replicate (Fig 1E and S1C Fig), demonstrating that EXP1 is essential for propagation in RBCs. The N-terminal fragment remaining after excision was nonfunctional as it does not rescue growth and was only detectable by IFA (S1D Fig), likely because of its small size and possible low stability.

Giemsa smears taken in regular intervals from synchronous parasites showed that ΔEXP1 ring stages were much slower to reach the trophozoite stage than controls (Fig 1F). ΔEXP1 trophozoites often displayed protrusions reaching into the host cell cytoplasm ("blebs," light blue arrowheads, Fig 1F) and frequently had an aberrant condensed morphology (Fig 1F). ΔEXP1 trophozoites did not complete schizogony as evident by a significantly reduced number of nuclei per ΔEXP1 parasite compared to controls (S1E Fig) and by an almost complete absence of new rings in the next cycle (Fig 1F). To test whether very slowly growing ΔEXP1 parasites persisted, we carried out an extended growth assay with the ΔEXP1 parasites. These experiments revealed a resurfacing of parasites in the ΔEXP1 culture grown in the presence of rapalog 9 days after loss of EXP1, but PCR identified this as a population with a nonexcised *exp1* locus. These parasites were therefore breakthroughs, further indicating that loss of EXP1 abolishes parasite propagation in RBCs (S1F Fig).

To observe the phenotype of EXP1 loss in more detail, we compared the development of ΔEXP1 parasites to controls, using long-term time-lapse imaging [21] (Fig 2A). This confirmed a severe delay of the ring stage (Fig 2B) and a very slow development of trophozoites without completion of the cycle (Fig 2A). Time-lapse imaging also revealed phenotypes in ring stages: ΔEXP1 parasites changed position in the host cell less frequently and rarely showed amoeboid shapes (Fig 2A and 2C), 2 typical features of ring stages [21] regularly observed in controls (Fig 2A and 2C). In addition, ΔEXP1 parasites were often found closely adjoined to the RBC membrane, a phenomenon that we here termed "hugging" (Fig 2A and 2C, red arrowheads), also evident by electron microscopy (S1G Fig).

The protrusions observed in Giemsa smears were also visible in differential interference contrast (DIC) and also in ΔEXP1 rings (Fig 2A, light blue arrowheads). These "blebs" were bounded by PPM and present almost exclusively in ΔEXP1 parasites as confirmed with a parasite plasma membrane (PPM) marker (Lyn-mCherry) [22] (Fig 2D). The total number of membrane-bounded protrusions detected using Bodipy-TR-ceramide or Lyso-PC was also mildly enriched in ΔEXP1 parasites (Fig 2E and S1H Fig). Despite these morphological alterations, the PVM integrity in ΔEXP1 parasites was not compromised, because a soluble PV marker (SP-mScarlet) was retained in the PV (S1I Fig).

Next, we evaluated the impact of EXP1 loss on gametocytogenesis. Interestingly, we detected stage III–V gametocytes lacking EXP1 (Fig 2F), although the number of late-stage gametocytes was reduced by more than 50% 8 days after induction (Fig 2G). Nevertheless, this indicated a lower effect of loss of EXP1 on the development of gametocytes than on asexual blood stages.

## All regions of EXP1 are important for its function

To confirm that the observed growth phenotype is specific for EXP1 loss, we complemented the condΔEXP1 parasites with a Ty-tagged full-length copy of EXP1 (EXP1wt-Ty) expressed under the constitutive *nmd3* promoter. EXP1wt-Ty was correctly located at the PVM (Fig 3A) and, after removal of the endogenous EXP1, restored parasite growth to 80% of the unexcised control (Fig 3B and S2A–S2C Fig). The level of complementation correlated with the level of expression of the EXP1wt-Ty construct, as demonstrated using promoters driving different levels of expression (Fig 3B–3D and S2A–S2C Fig). The complemented parasites also showed a similar duration of the ring stage compared to the wild type, indicating that the delay to reach the trophozoite was reverted by the complementation (S2D Fig).

To pinpoint the functional regions in EXP1, we tested a series of modified Ty-tagged EXP1 constructs for their capacity to complement loss of endogenous EXP1. Except for EXP1wt-mScarlet, all constructs were correctly inserted into the PVM, and deletions or replacements

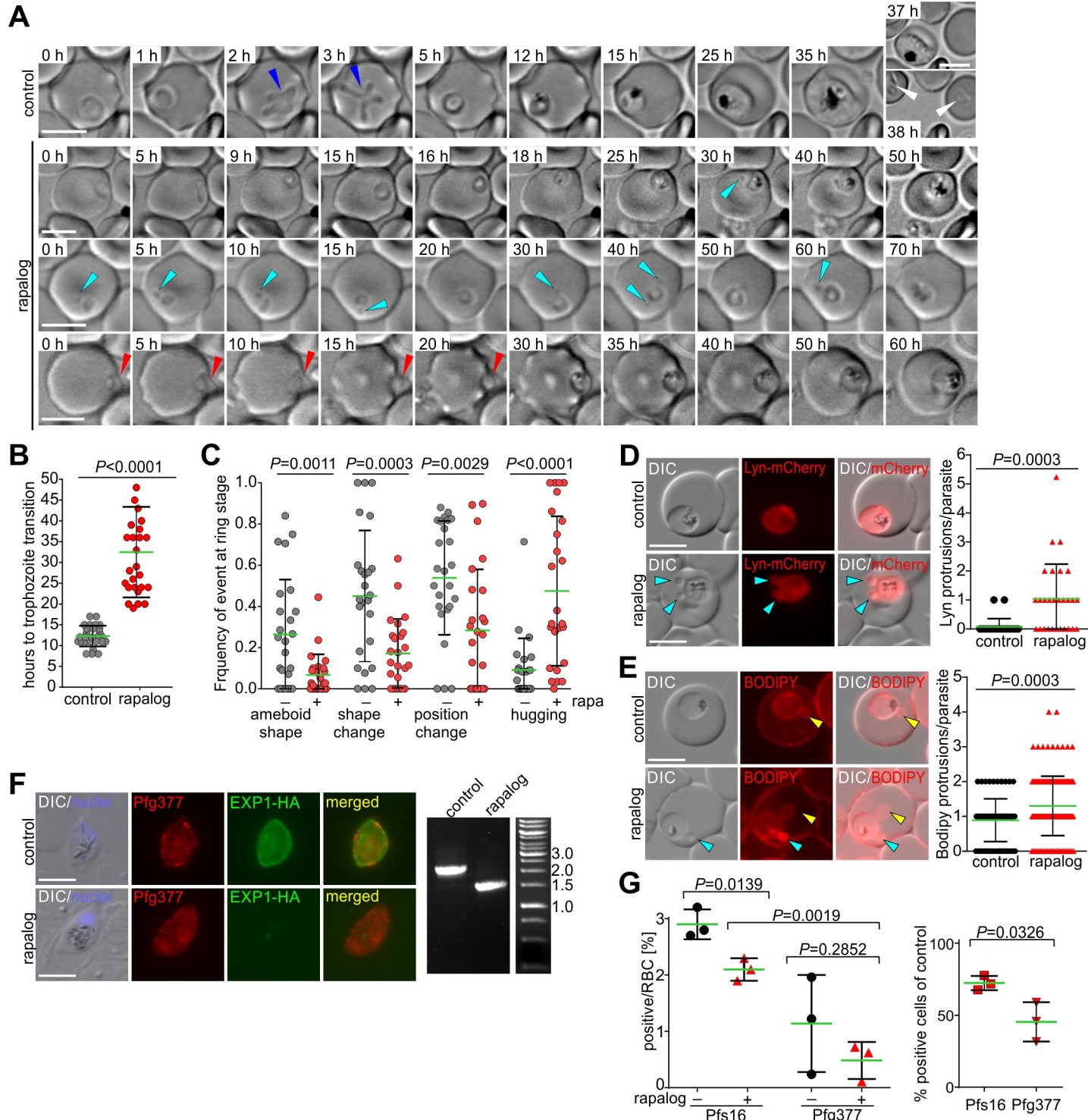

**Fig 2. Morphological phenotypes of ΔEXP1 parasites.** (A) Time-lapse imaging of condΔEXP1 parasites starting a cycle with (control) and without EXP1 (rapalog) imaged side by side. Single DIC z-sections of selected time points of typical phenotypes are shown: top row: slow development; mid row: failure to reach full trophozoite stage; bottom row: cell showing extensive hugging. Arrowheads: white, ring stages after reinvasion; blue, amoeboid ring; light blue, blebs; red, hugging. (B) Number of hours (after start of the time lapse experiment) to reach young trophozoite stage in *n* = 24 control and *N* = 27 ΔEXP1 parasites. (C) Frequency of the indicated events in individual ring stages observed by time lapse microscopy (presence or absence of event was scored every hour); *n* = 25 parasites each for ΔEXP1 and control. (D, E) Left, live cell images of control and ΔEXP1 (rapalog) parasites expressing Lyn-mCherry (PPM marker) (D) or labelled with Bodipy TR ceramide (BODIPY) (E). Light blue arrowheads, blebs; yellow arrowheads, TVN. Graphs: quantification of number of blebs per cell (controls *N* = 25, 73 for (D) and (E), respectively; ΔEXP1 (rapalog) *n* = 28

and 156 for (D) and (E), respectively. (F) Left, IFA of control and ΔEXP1 (rapalog) late-stage gametocytes using α-HA to detect EXP1*-HA and α-Pfg377 (late-stage gametocyte marker). Right, PCR using primers P1 and P2 (Fig 1A) confirms excision of *exp1* in late gametocytes. (G) Left, gametocytemia of control and ΔEXP1 parasites early after induction (Pfs16 positive cells) and 8 days later (Pfg377 positive cells) based on IFAs. Right, fold reduction in the number of early (Pfs16) and late (Pfg377) ΔEXP1 gametocytes versus control; *n* = 3. (D, E, and F); scale bars, 5 μm. In (F), nuclei were stained with DAPI. In (B–E and G), green lines indicate mean and error bars (SD); two-tailed unpaired *t* test, *P* values indicated. DIC, differential interference contrast; EXP1, exported protein 1; HA, triple hemagglutinin tag; IFA, immunofluorescence assay; PPM, parasite plasma membrane.

in the N- or C-terminus mostly led to severe loss of function (S2A-S2C and S3A and S3B Fig). Interestingly, EXP1 of the rodent malaria parasite *P. berghei* (*Pb*EXP1) only partially (46.1% ± 9.4% activity) rescued loss of EXP1 in *P. falciparum* (Fig 3B), indicating limited functional conservation between species. Deletion of the entire C-terminus of EXP1 reduced its activity to 59.37% ± 13.82% (Fig 3B). EXP1 lacking an 11 amino acid stretch (sequence SGVSSKKK NKK) in the N-terminus named E-domain (EXP1ΔED), a region proposed to be necessary for the dimerization based on similarity with MAPEGs [16,23], complemented only poorly (Fig 3B). Previous work indicated that EXP1 homo-oligomerizes [12]. However, loss of function of EXP1ΔED was not due to profound alterations in its capacity to oligomerize, as dimers were still detectable after formaldehyde crosslinking (Fig 3E).

We noticed that the TM domains of integral PVM proteins such as EXP1 and ETRAMPs in different malaria species are particularly rich in G, S, and T residues (S2E Fig). G, S, and T are known to be important as TM interaction interfaces [24, 25]. Strikingly, point mutations of the

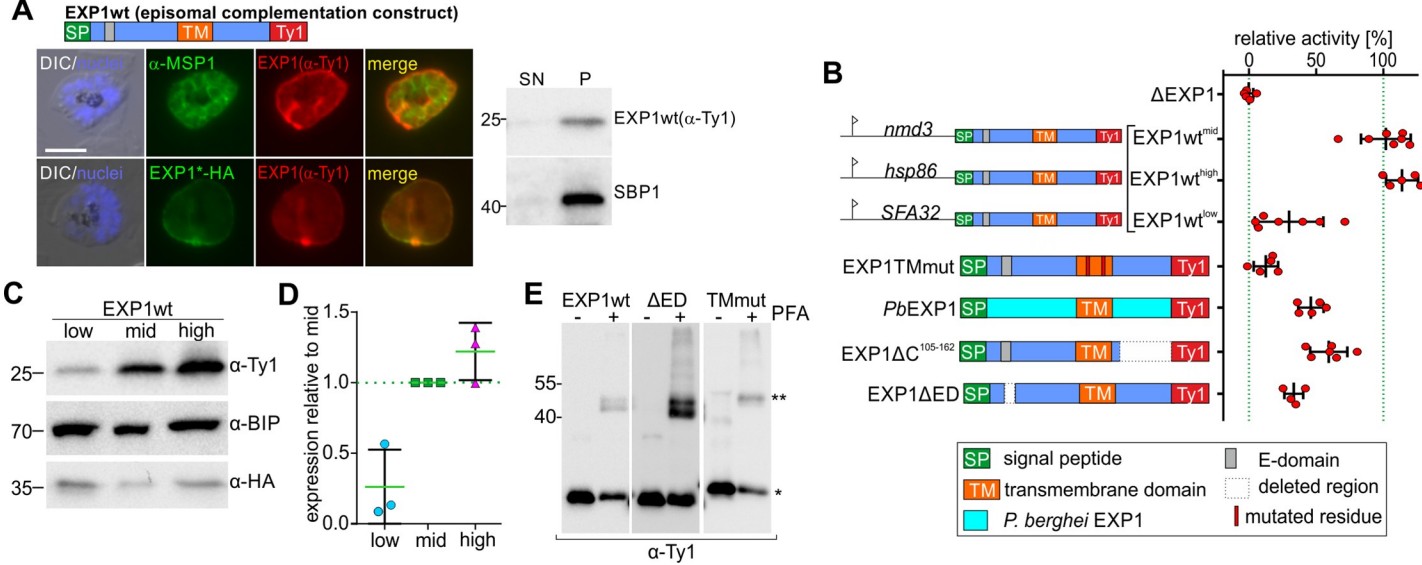

**Fig 3. Complementation pinpoints important regions of EXP1.** (A) IFA images of condΔEXP1 schizont stages expressing EXP1 wt-Ty probed with α-HA and α-Ty show localization of EXP1*-HA and EXP1 wt-Ty in the PVM. α-MSP1 (MSP1) labels the PPM. DAPI, nuclei; scale bars, 5 μm. Right: immunoblot of extracts of cell line on the left probed with α-Ty to detect EXP1wt-Ty and α-SBP1 as control for a TM protein. Saponin was used to separate the parasite pellet ("P") from the supernatant ("SN") containing PV and host cell content. See S3 Fig for IFAs and immunoblots of all complementation constructs. (B) Relative activity of the EXP1 complementation constructs indicated. Except where otherwise indicated, constructs were expressed under the *nmd3* (mid) promoter. The complementation capacity of every tested construct was calculated as relative activity to the EXP1wt-Ty construct under the *nmd3* promoter, which was set as 100% (right dotted green line). ΔEXP1 was set to 0% (left dotted green line). Each data point (red dot) shows growth of rapalog-treated versus unexcised parasites at the end of a 5-day growth assay relative to the growth of the wt construct; *n* ≥ 4 independent experiments per cell line. Error bars: SD. See S2 Fig for activity of all complementation constructs. (C) Immunoblot of lysates of condΔEXP1 parasites expressing EXP1low, EXP1mid, and EXP1high probed with α-Ty (EXP1wt-Ty), α-HA (EXP1*-HA), and α-BIP (loading control). (D) Densitometric analysis of EXP1wt expression levels (C) under low, mid, and high promoters relative to the mid promoter (green). Mean of 3 independent experiments. Error bars: SD. (E) Immunoblot of extracts of +/−formaldehyde (PFA) treated cell lines expressing the indicated constructs probed with α-Ty. Single asterisk: monomer; double asterisk: dimer. BIP, binding immunoglobulin protein; DIC, differential interference contrast; EXP1, exported protein 1; HA, triple hemagglutinin tag; IFA, immunofluorescence assay; MSP1, merozoite surface protein 1; PPM, parasite plasma membrane; PV, parasitophorous vacuole; PVM, parasitophorous vacuolar membrane; SBP1, skeleton binding protein 1; TM, transmembrane; wt, wild type.

first glycine residues (from G to L) of the two GXXG motifs of the *P. falciparum* EXP1 (*Pf*EXP1) TM abolished EXP1 function (Fig 3B) although the protein was still correctly trafficked (S3A and S3B Fig) and capable of dimerizing (Fig 3E). Collectively, we conclude that all parts of EXP1 are required for its function.

## EXP1 GST activity is dispensable for parasite growth

Previous work showed that recombinant EXP1 conjugates hematin to GSH in vitro and thereby may protect the parasite from heme-induced oxidative damage [16]. Based on homology with other MAPEGs, it was postulated that the catalytic center of the GST activity of EXP1 resides in 3 N-terminal residues (Fig 4A). Mutation of one of these residues (R70) led to a reduced enzymatic activity in vitro [16]. We used our complementation approach to assess the importance of these residues (and hence of the proposed GST activity) for parasite development. Constructs with mutations of 1 (EXP1R70A) or all 3 residues (EXP1-3xmut) of the proposed catalytic site showed 69.0% ± 9.2% and 62.5% ± 16.2% complementation activity, respectively (Fig 4B), indicating that these mutations had only a moderate effect on EXP1 function compared to most other complementation constructs (Fig 3B and S2A Fig). IFAs and solubility assays showed correct targeting of the complementation constructs to the PVM (S3A and S3B Fig). EXP1 R70A expressed under the stronger promoter resulted in nearly identical complementation capacity to the wild-type construct (Fig 4B). These data indicate that if EXP1 is a MAPEG, its GST activity plays only a minor role for growth of blood stage parasites.

## Loss of EXP1 is not associated with elevated oxidative stress

If EXP1 is a GST that protects from heme-induced oxidative damage, loss of its activity should lead to increased oxidative stress in the parasite [16]. To test whether EXP1, irrespective of our complementation data, may act as a GST in the parasite, we measured the oxidative stress status in ΔEXP1 and control age-matched trophozoites using the fluorescent reporter chloromethyl dihydrochloro fluorescein ($CM_2$-DCFDA) [26] (Fig 4C) to quantify intracellular reactive oxygen species (ROSs). Interestingly, the levels of fluorescence in ΔEXP1 parasites were not significantly different from those of control parasites (Fig 4D and S4A and S4B Fig), suggesting that ΔEXP1 parasites are not under elevated oxidative stress.

Consistently, ΔEXP1 parasites were not rescued in the presence of antioxidants (Trolox, ascorbic acid) and glutathione precursors (N-acetylcysteine and cysteine) (Fig 4E). Growth of parasites complemented with limiting amounts of EXP1 (EXP1wt$^{low}$) was also not ameliorated in the presence of any of the supplements (Fig 4E). Overall, these data indicate that the growth defect of ΔEXP1 parasites is not caused by elevated oxidative stress and reduction of oxidative damage does not rescue loss of EXP1.

The oxidative insult generated by hemoglobin byproducts can be diminished by inhibiting hemoglobin digestion with the cysteine protease inhibitor E64 [26, 27]. If EXP1 is involved in protecting the parasites from heme-mediated oxidative damage as proposed [16], growth of ΔEXP1 parasites should improve after inhibiting hemoglobin degradation. To test this, we treated ring stages expressing limiting amounts of EXP1 (EXP1wt$^{low}$) with E64 (before start of hemoglobin ingestion) and removed the inhibitor 12 hours later. Efficient inhibition of hemoglobin degradation was evident by swollen food vacuoles [28] (Fig 4F, arrowheads). However, E64 treatment did not restore growth of these parasites (Fig 4F). To confirm that E64 can protect against oxidative stress, we treated these parasites with dihydroartemisinin (DHA). While E64 ameliorated the effect of DHA as previously reported [29], E64 did not improve the growth defect of parasites expressing limiting levels of EXP1 (Fig 4G). Hence, the growth of parasites with reduced levels of EXP1 was not ameliorated by lower levels of hemoglobin-

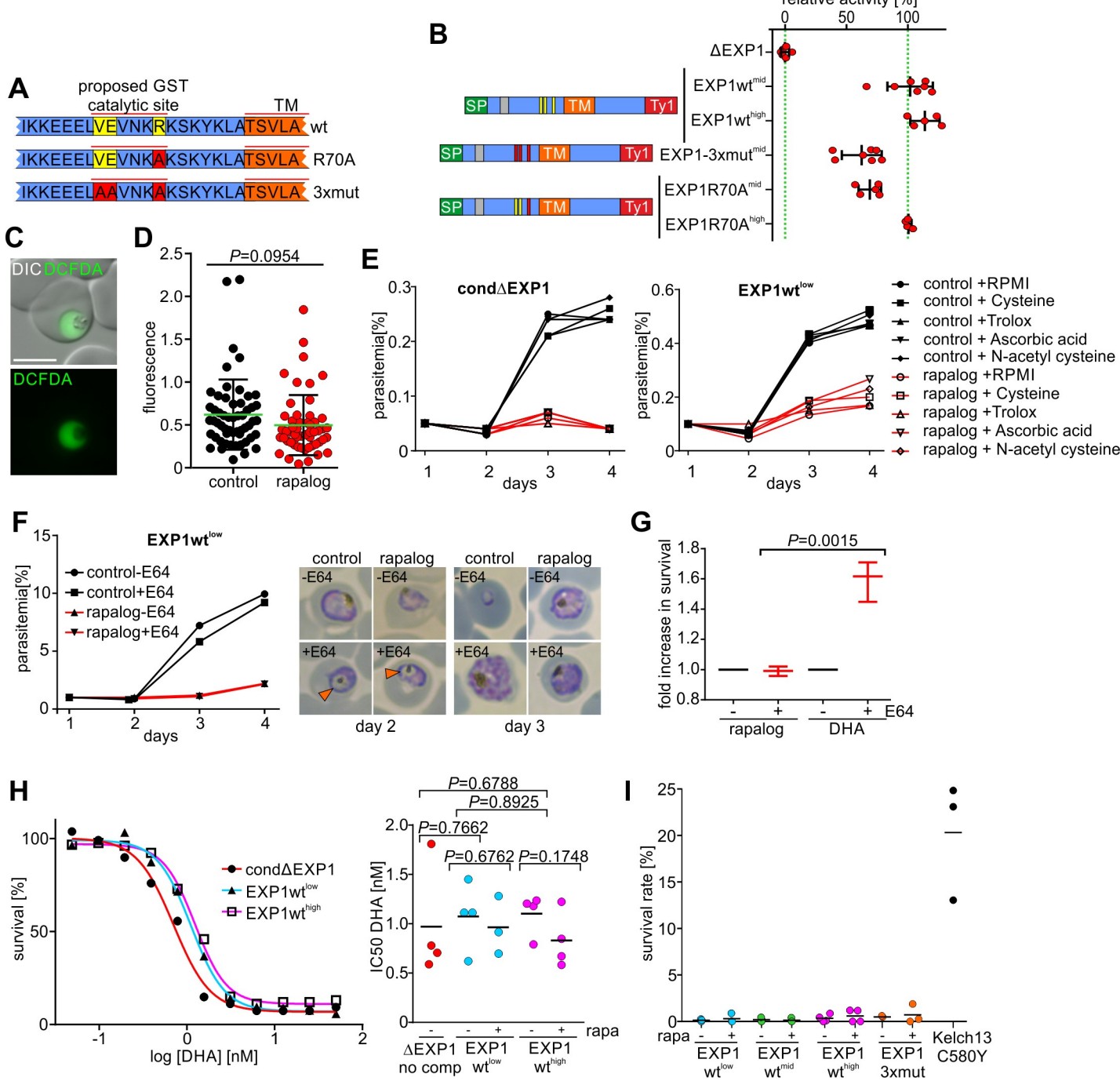

**Fig 4. Dispensability of EXP1 GST activity and lack of oxidative stress in ΔEXP1 parasites.** (A) Schematic of the region of EXP1 with the proposed catalytic site of the GST activity and of the mutations introduced. (B) Relative activity of the EXP1 complementation constructs indicated. Green lines: activity of EXP1wt (*nmd3*, mid) (set as 100%) and absence of activity (ΔEXP1) set as 0%; $n \geq 4$ independent experiments per cell line. (C) Live cell images of condΔEXP1 parasites incubated with CM-H$_2$DCFDA. Scale bars: 5 μm. (D) Fluorescence intensity of matching stages of control and ΔEXP1 parasites (rapalog) incubated with CM-H$_2$DCFDA (C). Results from 3 independent experiments with a total of $n = 55$ control and $n = 53$ ΔEXP1 cells. Green line, mean; error bars, SD. (E) FC growth curves of synchronous condΔEXP1 parasites and the complementation cell line EXP1 wt$^{low}$ after one cycle with (red) and without (black) rapalog (see Fig 1D) grown in RPMI alone or supplemented with the compounds indicated. One representative of $n = 3$ independent biological replicas. (F) Left, growth curves of EXP1wt$^{low}$ parasites ± E64 starting after one growth cycle ± rapalog (see Fig 1C). Right, Giemsa smears of parasites on day 2 (after incubation with E64) and day 3 (after removal). Arrowheads: swollen food vacuole. One representative of $n = 3$ biological replicas. (G) Effect of E64 treatment on survival of DHA and rapalog-treated EXP1wt$^{low}$ parasites versus untreated. Mean of $n = 3$ independent experiments. (H) Left, dose-response curves of the parasite lines indicated treated with DHA (0–50 nM). Right, DHA IC$_{50}$ values for these cell lines ± rapalog. Mean of $n \geq 3$ experiments. (I) RSAs of the indicated complementation cell lines and a Kelch13 C580Y mutant line. Data points are percent survival

of DHA treated versus untreated parasites ± rapalog. Mean of $n \geq 3$ per cell line. (D, G, and H), two-tailed unpaired $t$ test; $P$ values indicated; (B, D, and G), error bars, SD. CM-H$_2$DCFDA,; DHA, dihydroartemisinin; DIC, differential interference contrast; EXP1, exported protein 1; FC, flow cytometry; GST, glutathione S-transferase; IC$_{50}$, half maximal inhibitory concentration; RPMI, Roswell Park Memorial Institute; RSA, ring-stage survival assay; wt, wild type.

byproduct–induced oxidative stress, indicating that the proposed detoxification of hemoglobin metabolites is not a major function of EXP1.

## EXP1 does not influence ART resistance

EXP1 was proposed to be involved in ART resistance by conjugating ART to GSH and thereby lowering oxidative damage. ART resistance was also associated with up-regulation of EXP1 [16]. We exploited our conditional ΔEXP1 parasites to determine whether the expression levels of EXP1 influence the susceptibility of the parasites to DHA. The levels of EXP1 in the parasites used for these experiments ranged from very low and growth limiting (EXP1wt$^{low}$ in ΔEXP1 parasites) to overexpression (EXP1wt$^{high}$ on top of the endogenous EXP1-HA) (Fig 3B–3D). Determination of the half maximal inhibitory concentration (IC50) for DHA showed no significant difference between these cell lines after removal of the endogenous EXP1 (Fig 4H). Naturally occurring ART resistance can only be measured using ring-stage survival assays (RSAs) [30]. To detect lower levels of resistance, we used a lower than usual concentration of DHA (350 nM). While a previously established DHA-resistant line [20] displayed reduced susceptibility to DHA in the RSA, the parasites expressing higher levels of EXP1 than wild type displayed no better survival than parasites expressing limiting levels of EXP1 or the GST catalytic site mutant EXP1 (Fig 4I), indicating that EXP1 levels did not affect ART responsiveness.

## The nutrient-permeable channel activity but not PTEX function is defective in ΔEXP1 parasites

As the previously postulated function as a heme-detoxifying GST did not appear to be responsible for the phenotype in ΔEXP1 parasites, we looked for other possible functions of EXP1. In previous work, we identified EXP1 in immunoprecipitations (IPs) of EXP2 [31]. Therefore, EXP1 might be involved in functions attributed to EXP2, either protein export or the nutrient-permeable channel activity at the PVM. First, we tested whether ablation of EXP1 affects protein export. ΔEXP1 parasites showed no defect in the export of skeleton binding protein 1 (SBP1), ring exported protein 1 (REX1), REX2 (early-stage exported), and knob-associated histidine-rich protein (KAHRP) and MSP7-related protein 6 (MSRP6) (late-stage exported) (Fig 5A and 5B and S4C Fig). Thus, PTEX is still functional in ΔEXP1 parasites.

To test whether EXP1 affects the nutrient-permeable channel activity at the PVM, patch-clamp measurements were performed on the PVM of ΔEXP1 parasites liberated from their host cell. After liberation, the PVM remained intact, as evidenced by the retention of a co-expressed soluble PV marker (Fig 5C and S1I Fig). The PVM of the liberated parasites was now accessible to a patch-clamp pipette. After giga-seal formation, each individual sample was inspected for channel activity, defined as a current flicker from closing channels at 30 mV applied voltage to the patch pipette [7]. While channel activity was often immediately apparent in the control sample, channel activity was mostly absent in the ΔEXP1 parasites (Fig 5D left). The frequency to detect at least one channel at the PVM (f$_{chan}$) of ΔEXP1 parasites was significantly reduced compared to controls (Fig 5D right). Together, these results demonstrate that EXP1 is important for the nutrient-permeable channel activity at the PVM but not for protein export.

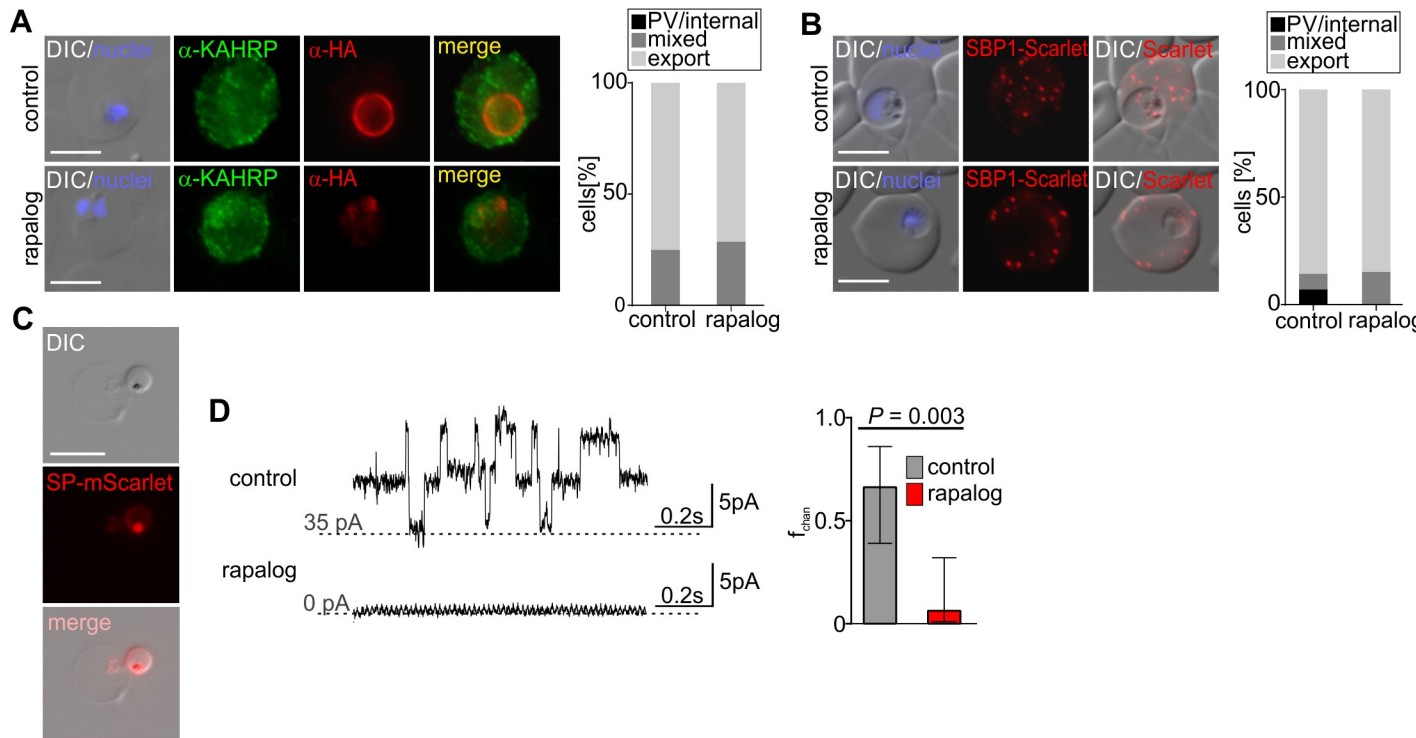

**Fig 5. PVM nutrient-permeable channel but not protein export is impaired in ΔEXP1 parasites.** (A) IFA images of control and ΔEXP1 parasites (rapalog) probed with α-HA (EXP1*-HA) and α-KAHRP. Graph to the right shows quantification of export (n = 20 cells). See S5C Fig for other exported proteins. (B) Live cell imaging of control and ΔEXP1 parasites (rapalog) expressing SBP1-mScarlet. Graph to the right shows quantification of export (n = 20 cells). (C) Live cell images of a ΔEXP1 trophozoite expressing SP-mScarlet liberated from its host RBC. (D) Left, current recorded from liberated control and ΔEXP1 parasites (rapalog) at 30 mV applied potential to the pipette electrode. Scale bar shows time in seconds and current in pA. The dotted line indicates a current reference level. At 30 mV, the PVM channels have an open probably of about one-half, therefore the flicker is offset in the control example as multiple channels are in the recording. The shown recordings are representative of the experiments done in each condition and show 1-second details from longer recordings to resolve the typical channel flicker in print. Right, probability of detecting at least one PVM channel per sealed patch ($f_{chan}$) in ΔEXP1 parasites (rapalog, n = 14) and controls (n = 12). Fischer's exact test was used to estimate P value. Error bars indicate SD. In (A) and (B), nuclei were stained with DAPI; scale bars: 5 μm. DIC, differential interference contrast; EXP1, exported protein 1; HA, triple hemagglutinin tag; KAHRP, knob-associated histidine-rich protein; pA, Picoampere; PSAC, parasite surface anion channel; PVM, parasitophorous vacuolar membrane; RBC, red blood cell; SBP1, skeleton binding protein 1.

## Loss of EXP1 alters the distribution of EXP2, and both proteins interact at the PVM

To investigate the effect of EXP1 loss on EXP2 localization as a possible reason for the reduction of nutrient-permeable channel activity, we expressed a green fluorescent protein (GFP) fusion of EXP2 (EXP2GFP) in condΔEXP1 parasites. No differences were obvious between controls and ΔEXP1 ring stages (S5A and S5B Fig). However, at the beginning of the trophozoite stage, EXP2-GFP displayed a profoundly altered distribution in 75% of the ΔEXP1 cells (Fig 6A–6C) as evident by the concentration of EXP2 in small regions of the PVM, frequently in loop-like Bodipy-TR-ceramide positive protrusions (Fig 6A and 6C). In control cells, EXP2-GFP was predominantly found in a circular pattern around the parasite (Fig 6A–6C). IFAs with specific antibodies confirmed this phenotype and showed an altered distribution of the endogenous EXP2 in ΔEXP1 parasites (Fig 6D and 6E and S5B Fig).

The distribution of other PVM proteins such as early (ETRAMP10.1) or later (ETRAMP4) integral PVM markers [10] showed no apparent differences in ΔEXP1 parasites compared to controls (Fig 6F and 6G). In contrast, ETRAMP5 accumulated in small regions of the PVM where it colocalized with the aberrantly distributed EXP2-GFP (Fig 6H). Interestingly,

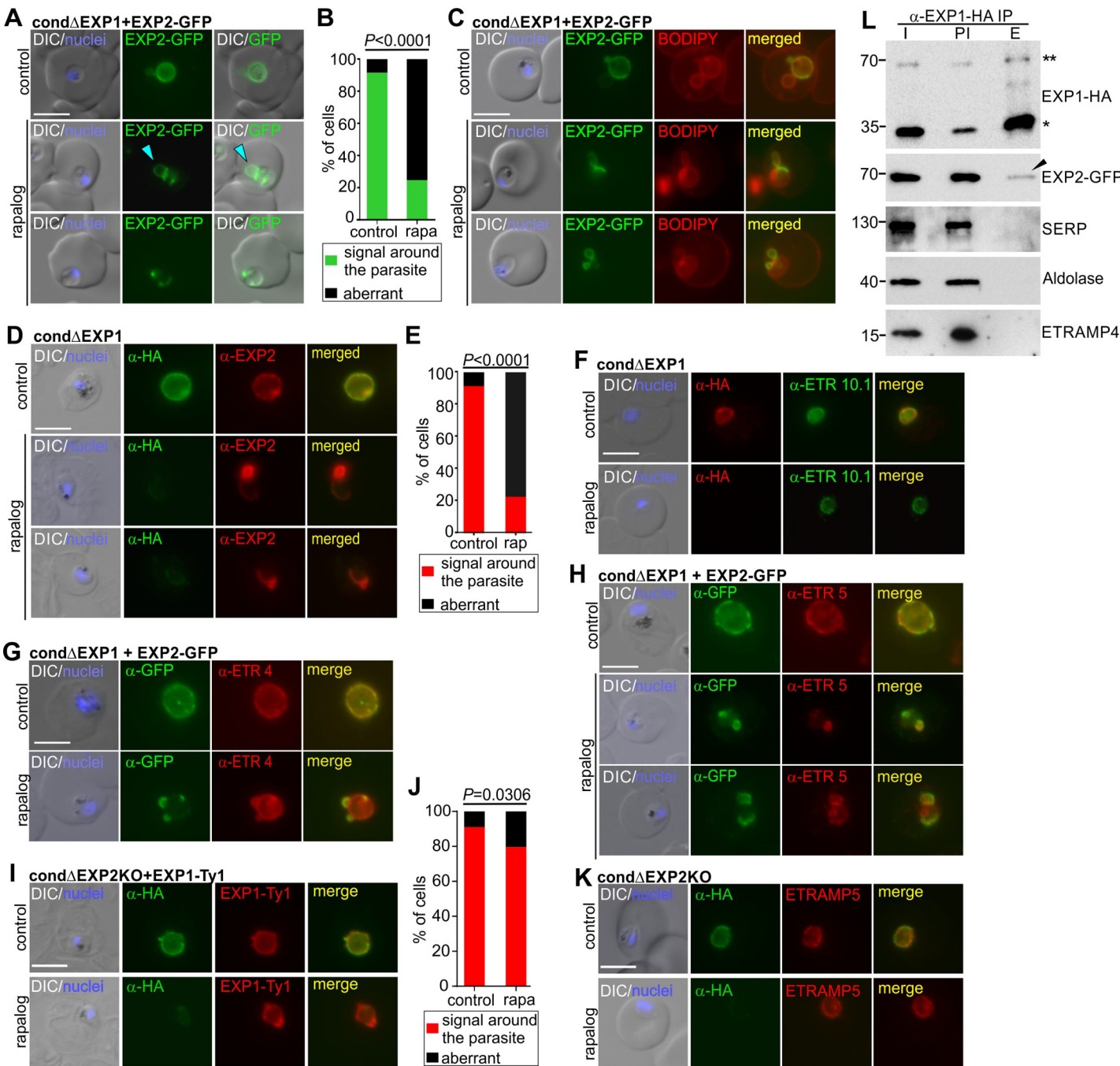

**Fig 6. Localization of EXP2 and PVM proteins in ΔEXP1 parasites and interaction analysis.** (A) Live cell images of ΔEXP1 (rapalog) and control trophozoites expressing EXP2-GFP. Light blue arrowhead, loop-like protrusions. (B) Quantification of the phenotype of the cells from (A). Green, signal around the parasite; black, aberrant distribution of signal. Mean of a total of $n = 109$ control and $n = 99$ ΔEXP1 cells derived from 4 biological replicas. (C) Live cell images of Bodipy-TR-ceramide labelled ΔEXP1 (rapalog) and control parasites expressing EXP2-GFP. (D) IFA images of ΔEXP1 (rapalog) and control parasites probed with α-HA to detect EXP1*-HA and α-EXP2 for endogenous EXP2. (E) Quantification of the phenotype of the cells from (D). Red, signal around the parasite; black, aberrant distribution of signal. Mean of a total of $N = 108$ control and $N = 135$ ΔEXP1 cells derived from 3 biological replicas. (F–H) IFA images of ΔEXP1 (rapalog) and control ring stages (F) or trophozoites (G, H) probed with α-HA (EXP1*-HA) and α-ETRAMP10.1 (F), α-GFP (EXP2-GFP) with α-ETRAMP4 (G), or with α-ETRAMP5 (H). (I) IFA images of ΔEXP2 (rapalog) and control trophozoites expressing EXP1-Ty probed with α-Ty and α-HA to detect EXP1-Ty and EXP2-HA, respectively. (J) Quantification of phenotypes of the parasites shown in (I). Red, signal around the parasite; black, aberrant distribution of signal. Mean of a total of $n = 86$ control and $n = 110$ ΔEXP1 cells derived from 3 biological replicas. (K) IFA images of ΔEXP2 (rapalog) and control parasites probed with α-HA and α-ETRAMP5. In (I) and (K), α-HA detects full (control) or truncated (rapalog) EXP2-HA. In (A, C, D, F–I, K), scale bar: 5 μm. DAPI, nuclei. (l) Western blot of a co-IP experiment in the cell line condΔEXP1 expressing EXP2-GFP (IP of EXP1*-HA with α-HA). α-HA detects EXP1*-HA (monomer: asterisk; dimer: double asterisk); α-GFP, EXP2-GFP (arrowhead); α-SERP, soluble PV protein; α-ETRAMP4, integral PVM protein; α-aldolase, cytosolic parasite protein. Input (I): total lysate before IP; post IP lysate (PI); Eluate (E). One

representative of *n* = 3 independent biological replicas. In (B, E, and J), *P* values were calculated with a Fischer's exact test. *P* < 0.05, significant. ETRAMP, early transcribed membrane protein; EXP1, exported protein 1; GFP, green fluorescent protein; HA, triple hemagglutinin tag; IFA, immunofluorescence assay; IP, immunoprecipitation; PV, parsitophorous vacuole; PVM, parasitophorous vacuolar membrane; SERP, serine-rich antigen also known as serine repeat antigen 5 (SERA5).

ETRAMP5 was previously found in co-IPs as a potential interaction partner of EXP2 [31]. However, in contrast to EXP1, ETRAMP5 was dispensable for parasite growth (S6 Fig) and may not be needed for an essential process such as the nutrient-permeable channel activity at the PVM.

As EXP1 influenced the distribution of EXP2, we tested whether this effect was reciprocal by knocking out EXP2. For this, we used the same strategy as for EXP1 to generate a conditional EXP2 KO (S7 Fig). As previously published, KO of EXP2 led to loss of protein export and arrested development at the trophozoite stage [7,32]. However, neither the distribution of EXP1-Ty expressed in these parasites nor that of endogenous ETRAMP5 was markedly altered in ΔEXP2 parasites (Fig 6I–6K). The minor effect observed on the distribution of EXP1 likely is due to the impact on parasite morphology in the EXP2 KO. Hence, the correct localization of EXP1 does not appear to depend on EXP2.

Prompted by the altered distribution of EXP2 in the ΔEXP1 parasites, we performed co-IPs in crosslinked parasites to test whether the two proteins interact. IP of the endogenously HA-tagged EXP1 in condΔEXP1 parasites expressing EXP2-GFP resulted in copurification of EXP2-GFP (Fig 6L), while a soluble PV protein (serine-rich antigen 5 [SERA5]) and an integral PVM protein (ETRAMP4) were not co-immunoprecipitated. The reciprocal experiment by immunoprecipitating EXP2-GFP corroborated these findings (S8A Fig). Thus, EXP1 interacts with EXP2 at the PVM, suggesting that their activity may be linked. We also found EXP1 and EXP2 in structures within merozoites (S8B Fig), in agreement with previous results showing that both proteins are stored in dense granules [33, 34].

Finally, we tested whether loss of EXP1 affected membrane association of EXP2. In the EXP1 KO parasites, EXP2 was still membrane associated as indicated by retention of the protein after lysing parasites with saponin (S8C Fig). Next, we carried out more detailed tests using carbonate and urea to solubilize peripheral membrane proteins. Previous work showed that a small fraction of EXP2 can be extracted with carbonate [34]. This property was not changed after removal of EXP1 (S8D Fig). While the extractability of EXP2 by urea appeared to be increased in the ΔEXP1 parasites compared to the control, this was not significant (S8D Fig). This indicated that loss of EXP1 does not profoundly alter the membrane association of EXP2.

## Levels of EXP1 influence the capacity of parasites to respond to amino acid starvation

Due to its additional role in protein export, knocking out EXP2 precludes specific analysis of the PVM nutrient-permeable channel function. In contrast, EXP1 loss only affected this activity (Fig 5D). In agreement with a role in nutrient uptake, the phenotype observed in the ΔEXP1 parasites (condensed trophozoites, growth retardation, and blebbing) resembled starvation phenotypes caused by amino acid depletion [35] and by the loss of the parasite surface anion channel (PSAC) [36,37], the activity at the RBC membrane for uptake of nutrients from the serum [38]. To first confirm that the phenotype of ΔEXP1 parasites was solely due to loss of the channel activity at the PVM, not at the RBC membrane, we assessed the uptake of 5-aminolevulinic acid (5-ALA) [39] into ΔEXP1 infected RBCs, an indicator for PSAC activity. Uptake of 5-ALA into RBCs infected with ΔEXP1 trophozoites was not significantly different

from controls (Fig 7A and 7B). The small trend for reduced uptake was likely due to the growth retardation of ΔEXP1 parasites (S8E Fig). Thus, PSAC activity is not impaired in ΔEXP1 parasites, and the starvation-like phenotype is caused by the loss of the nutrient permeability at the PVM.

To more specifically test the association of EXP1 with nutrient acquisition, we compared parasite growth in medium containing limiting concentrations of amino acids with parasites grown in complete medium in cell lines expressing different levels of EXP1. Interestingly, growth in limited medium mimicked the prolonged ring-phase phenotype observed in ΔEXP1 parasites (Fig 7C and 7D). Furthermore, while all cell lines expressing physiological (or higher) levels of EXP1 tolerated the limiting medium over 2 growth cycles to a similar extent, the cell line expressing limiting levels (EXP1$^{low}$ on rapalog) was hypersensitive to low levels of amino acids (Fig 7E). This hypersensitivity was specifically related to the lack of amino acids because its response to an unrelated growth inhibition (using sodium azide) was similar to the other cell lines (Fig 7E). We conclude that EXP1 is critical for nutrient acquisition across the PVM and that the presence of the nutrient-permeable channel detected by patch clamping at the PVM correlates with nutrient uptake. Parasites relying only on EXP1 with the mutated catalytic site showed an intermediate sensitivity to the limited medium, indicating that the growth reduction in these parasites (Fig 4B) was also due to reduced nutrient acquisition with this version of EXP1, although there was no significant difference to the line expressing both, the mutated and the wild-type form of EXP1 (Fig 7E).

## Discussion

Previous work led to the proposal that EXP1 is a GST of the MAPEG family that detoxifies hemoglobin byproducts and thereby protects malaria parasites from oxidative stress [16]. Furthermore, this work indicated that through this GST function, EXP1 can protect the parasite from the action of ART and that EXP1 transcription levels were elevated in ART-resistant parasites. Here, we show that the postulated GST activity of EXP1 is largely dispensable for parasite survival and that EXP1 protein levels did not influence ART susceptibility. Inhibition of hemoglobin catabolism by a protease inhibitor or the supplementation with reducing agents to protect from oxidative damage did not improve parasite growth when EXP1 was absent or its expression levels were growth limiting. Hence, although EXP1 may have GST activity in vitro, overall, our data indicate that this activity is not critical for blood stage growth. MAPEGs are a highly diverse and widely distributed family of proteins involved in the detoxification of metabolites and in glutathione and lipid metabolism [15]. It is possible that EXP1 derives from such proteins but has adopted other functions that are critical for parasite survival.

We here provide functional evidence for a different role of EXP1 that is important for parasite survival. Apicomplexan parasites require an external supply of nutrients that reach the parasite by passive diffusion through a nonselective pore of the PVM [6]. Our patch-clamp experiments indicate that EXP1 is essential for the activity of a nutrient-permeable channel in the PVM. Recently, a different PVM protein, EXP2, was shown to be needed for this activity [7,9]. EXP2 oligomers also form the membrane-spanning pore of PTEX [3,8]. Hence, EXP2 is needed for both, the nutrient-permeable channel activity and for protein export, potentially as the pore of both activities [7]. In previous work, Gold and colleagues postulated that EXP2 is either part of a complex that mediates both, protein transport and the nutrient-permeable channel function or that it is part of 2 molecularly distinct complexes, each serving only one of these functions [9]. Recent data showed that there is a large pool of EXP2 in trophozoites that is devoid of the PTEX component heat shock protein 101 (HSP101), lending support for the existence of 2 compositionally distinct EXP2 complexes [7]. However, targeting EXP2 cannot

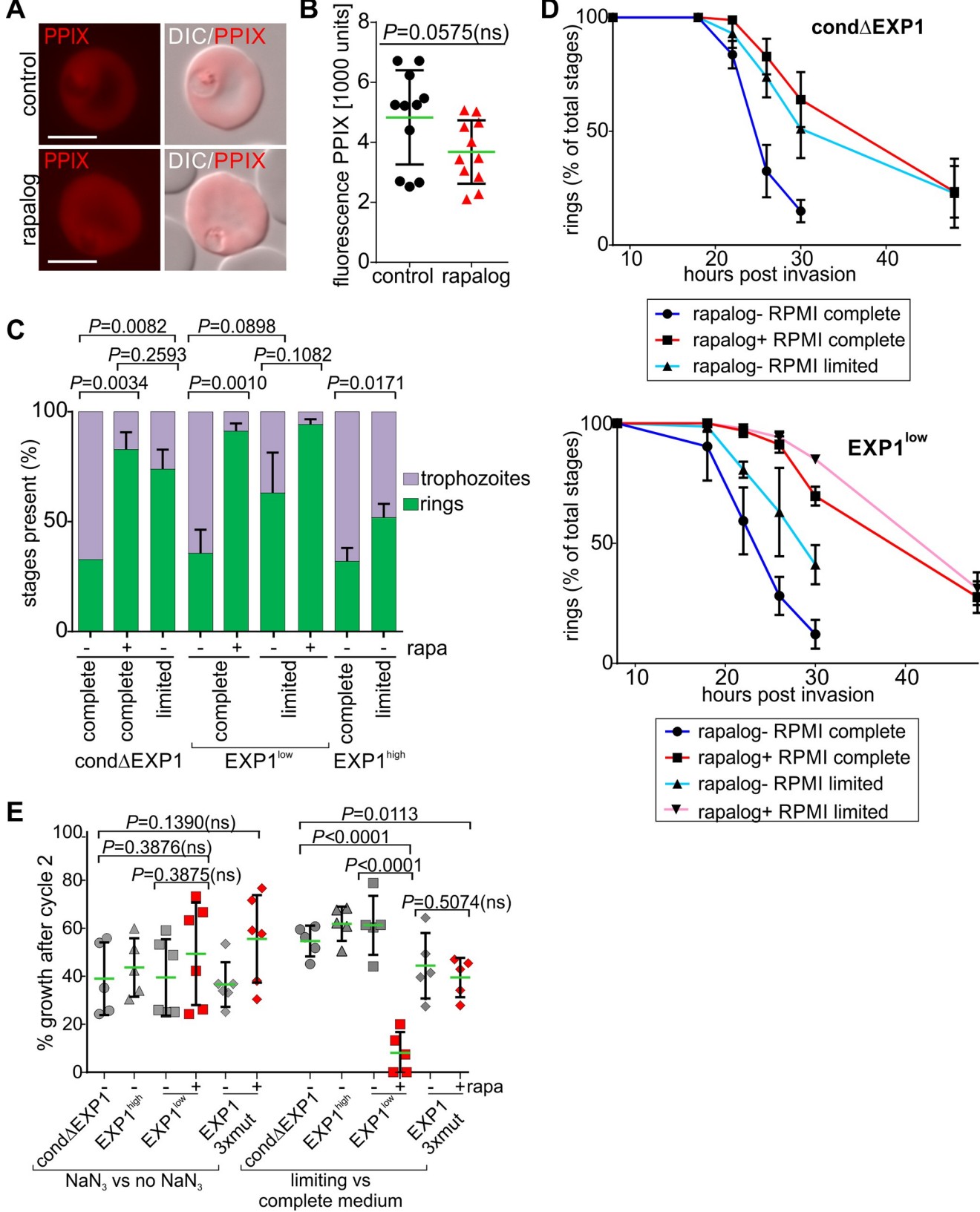

**Fig 7. Reduced levels of EXP1 mimic starvation and lead to hypersensitivity to low nutrients.** (A) Live cell images of control and ΔEXP1 parasites (rapalog) incubated with 5-ALA. (B) Quantification of PPIX fluorescence in control and ΔEXP1 parasites (rapalog) using FC. Mean (green line) of $n = 11$ independent experiments. Error bars indicate SD. (C) Stage distribution of tightly synchronous parasite cell lines 24 h.p.i. (after an initial cycle ± rapalog) grown in amino acid–limited and complete medium. Mean of $n = 3$ independent experiments. (D) Percentage of rings at different time points post invasion (after 1 cycle ± rapalog) of condΔEXP1 and EXP1wt$^{low}$ parasites grown in amino acid–limited and complete medium. Mean of $n = 3$ independent experiments. (E) Growth on day 4 (after 2 cycles) of the indicated parasite lines and condition (±rapalog) in the presence of azide in proportion to growth of the same parasites in medium without azide (left, $NaN_3$ versus no $NaN_3$) or growth in amino acid–limited medium in proportion to the same parasites grown in complete medium (right, limiting versus complete medium). Growth of the control culture (medium without $NaN_3$ or complete medium) was set as 100%. Rapalog was added 1 cycle prior to the growth test to start with the corresponding KO parasites. Green line indicates mean of at least $n = 5$ independent experiments. In (B, C, and E), two-tailed unpaired $t$ test, $P$ values are indicated. 5-ALA, 5-aminolevulinic acid; DIC, differential interference contrast; EXP1, exported protein 1; FC, flow cytometry; h.p.i., hours post invasion; KO, knockout; PPIX, protoporphyrin IX; rapa, rapalog; wt, wild type.

functionally distinguish these complexes. We here found that knocking out *exp1* specifically affects the location of EXP2 in trophozoites and abolished the nutrient-permeable channel activity at the PVM but not protein export. We also show that EXP1 interacts with EXP2. Hence, EXP1 is a defining factor of this nutrient-permeable channel activity, likely through its interaction with EXP2 and by maintaining the correct localization of this protein. It is therefore likely that there are indeed 2 functionally and compositionally distinct pools of EXP2, one serving protein export (the PTEX complex) and one for the nutrient-permeable channel function (depending on EXP2 and EXP1).

In further agreement with a PTEX-independent pool of EXP2, expression of EXP2 peaks in trophozoites, not in rings like the other PTEX components [7,34]. Interestingly, EXP1 has a very similar expression profile to EXP2 [40] (Fig 8A), congruent with a function together with the trophozoite expressed EXP2 in the nutrient-permeable channel activity (Fig 8B). Overall, this indicates that in rings, when protein export is a predominant requirement for the parasite, EXP2 exists foremost in PTEX. In trophozoites, when protein export is less important but nutrient acquisition for rapid growth is critical, most of EXP2 is needed together with EXP1 for nutrient uptake and derives from a pool expressed after production of PTEX. As the PTEX components (apart from EXP2) are not, or only poorly, expressed in trophozoites, residual PTEX complexes remaining from the ring stage are likely sufficient to accomplish the protein export needs in trophozoites. Hence, PTEX likely occupies only a minority of the EXP2 population in trophozoites (Fig 8B). This is supported by our finding that knocking out EXP1 specifically affected the location of EXP2 in trophozoites but not in rings. Overall, the current data support a model in which protein export and the nutrient-permeable channel function are accomplished by molecularly distinct complexes that share EXP2 and have reciprocal stage-specific expression peaks that temporally coincide with the needs of the respective parasite stages (Fig 8).

How loss of EXP1 affects the function of the nutrient-permeable channel activity remains to be determined. One possibility is that EXP1 has a role as part of the nutrient pore structure, and its absence could directly lead to a defective channel (Fig 8B). Alternatively, EXP1 may have a more indirect role. In this respect, it is interesting to note that EXP1 and the topologically related ETRAMPs form oligomeric arrays in the PVM and were hypothesized to compartmentalize different activities of the PVM [12]. The mislocalization of EXP2 could indicate that the EXP1 KO leads to EXP2 aggregation or otherwise negatively affects a functionally relevant spatial distribution of this protein in the membrane (Fig 8B). This effect could be specific for EXP2, as suggested by our IP data and the fact that ETRAMP4 did not show an altered distribution in the PVM of ΔEXP1 parasites. A more general effect on the state of PVM proteins cannot not be fully excluded. The EXP1 KO also affected the distribution of ETRAMP5 in the PVM. However, this could also mean that this ETRAMP is connected to EXP2 and EXP1. This possibility is supported by the fact that ETRAMP5 also appeared in the EXP2 interactome [31]

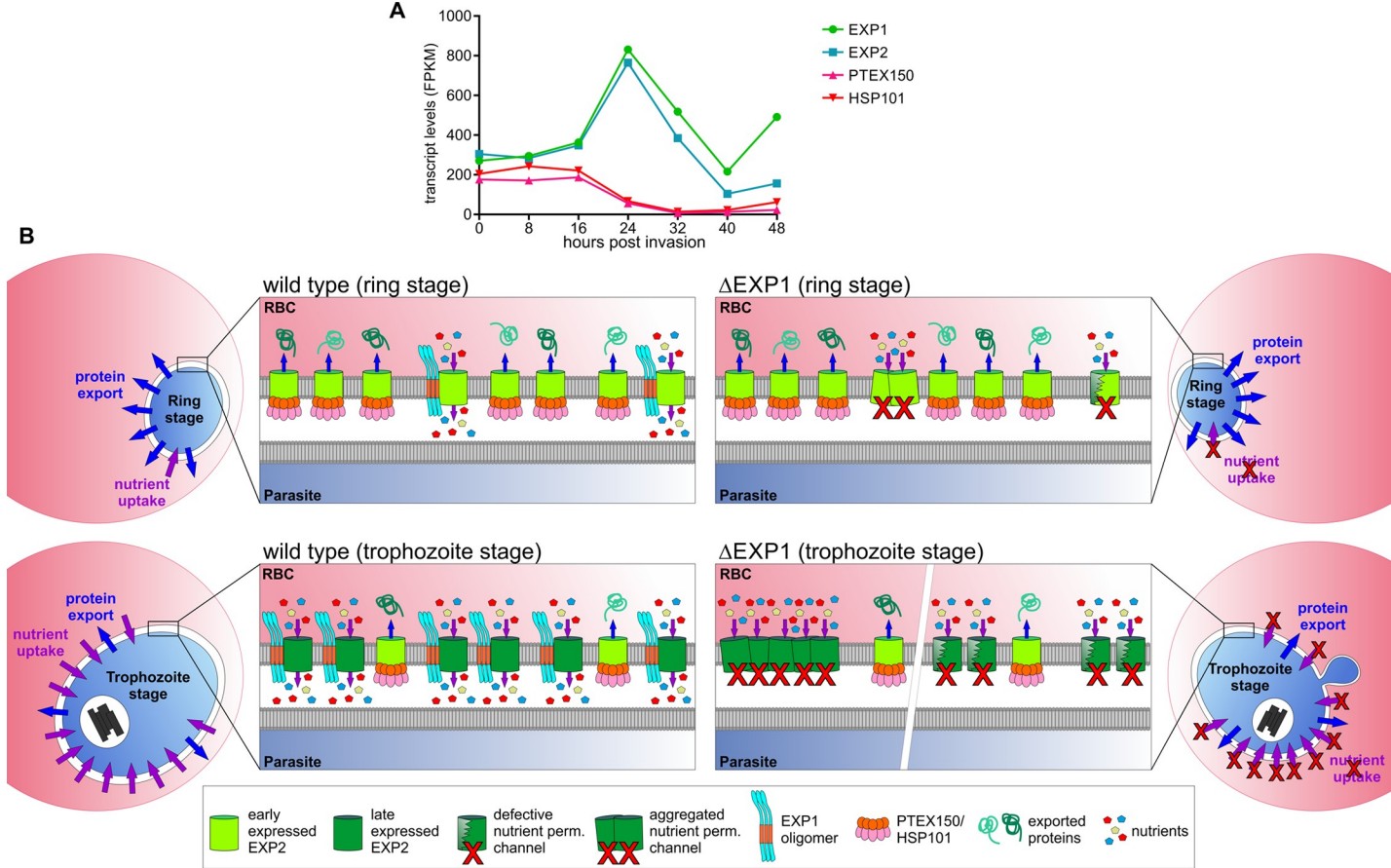

**Fig 8. Model of the EXP1-defined nutrient-permeable channel function of EXP2 distinct from the PTEX complex.** (A) Transcript levels of EXP1, EXP2, and the PTEX components HSP101 and PTEX150 across the asexual intra-erythrocytic cycle. Values were obtained from a previous publication [40]. (B) Schematics of the PVM of wild-type (left) and ΔEXP1 parasites (right) during the ring (upper panel) and trophozoite stage (lower panel). Depicted molecules are explained in the box. Note that early expressed EXP2 is in light green and late expressed EXP2 is in dark green. In wild-type rings, EXP2 is predominantly associated with PTEX components to promote protein export, a major function of this stage. Less EXP2 is in the nutrient-permeable channel complex that depends on EXP1 but does not contain other PTEX components. The proportion of the abundance of the two EXP2 complexes is reversed in the trophozoite stage when most of protein export has been completed but rapid parasite growth demands more nutrients. PTEX150 and HSP101 are not expressed in this stage, but PTEX translocons left over from the ring stage make up a minor proportion of EXP2 complexes to maintain protein export activity. The nutrient-permeable channel function of EXP2 is negatively affected by loss of EXP1 either through indirect effects (indicated as aggregated EXP2, left part of PVM enlargement of ΔEXP1 KO) or a direct defect of the complex ("defective" nutrient-permeable channel, right part of ΔEXP1 KO). EXP1, exported protein 1; FPKM, fragments per kilobase of exon model per million mapped reads; h.p.i., hours post invasion; HSP101, heat shock protein 101; KO, knockout; PTEX, *Plasmodium* translocon of exported proteins; PVM, parasitophorous vacuolar membrane.

and shows a similar expression profile to EXP1 and EXP2 [40]. Nevertheless, if this is the case, ETRAMP5 does not play a critical part, as we here show that it is not essential for growth of asexual blood stages.

Consistent with a function in nutrient uptake of the parasite, EXP1 loss resembled previously observed starvation phenotypes [35, 36, 37], and EXP1 levels specifically influenced the capacity of parasites to grow in medium with low levels of amino acids. This supports a role of EXP1 in nutrient acquisition and indicates that the activity measured at the PVM indeed is critical for the parasite's nutrient supply. Such a role for EXP1 now allows us to specifically study the nutrient-permeable channel activity, independent from the protein export activity of EXP2.

We confirmed previous data that EXP1 and EXP2 are stored in merozoites [33,34], indicating that not only PTEX but also the nutrient-permeable channel activity can be delivered to

the PVM during invasion and thus be active in ring stages. This would allow access of nutrients to rings and would explain the delayed development of these stages in the EXP1 KO. Nevertheless, most ΔEXP1 rings eventually reached the young trophozoite stage, likely reflecting the lower dependence on exogenous nutrients up to this stage. The resemblance of a nutrient starvation phenotype did not derive from a restricted nutrient access to the infected RBC but rather is consistent with a loss of the nutrient permeability at the PVM in agreement with our patch-clamp observations. It is also noteworthy that the effect of the EXP1 KO was less pronounced on gametocytes, which are less metabolically active during their later development [41].

In conclusion, EXP1 is critical for the replication of *P. falciparum* parasites in RBCs, but this is independent of its postulated heme-detoxifying membrane GST activity. Rather, the evidence presented in this study supports the thesis that EXP1 is required for the EXP2-based nutrient-permeable channel activity of the PVM. Accordingly, the activity measured by patch-clamp indeed is important for the access of metabolites needed for parasite replication. These data clearly show that the function of EXP2 for nutrient uptake is functionally distinct from its role in protein transport.

## Materials and methods

### Plasmid constructs cloning

For SLI-targeted gene disruption, the first 575 bp of the *exp1* gene and the first 291 bp of the *etramp5* gene without the start codon (including introns) were cloned into pSLI-TGD [20] using NotI and MluI (oligonucleotides and the plasmid constructs are listed in S1 Table). For conditional deletion of *exp1*, the first 575 bp of the *exp1* gene were PCR amplified using primers to append a double myc tag, a first loxP site (not disrupting the open reading frame) and a recodonized T2A skip peptide. A codon changed *exp1* gene was synthesized (Genscript) and PCR amplified with primers to add a second loxP site after the gene to obtain a second fragment. Both fragments were cloned into pSLI-3xHA, (derived from pSLI [20] by replacing GFP with 3xHA) using NotI and KpnI by Gibson cloning. This resulted in plasmid pSLI-*exp1-loxP*. Using the same strategy for the conditional excision of *exp1*, the first 986 bp (including introns) of the *exp2* gene were amplified using primers to add a first loxP site and a recodonized T2A skip peptide to obtain a first fragment. A recodonized *exp2* gene was synthesized (Genscript) and amplified with overlapping primers to append a second loxP site. Both PCR products were cloned into pSLI-*exp1-loxP* using NotI and XmaI by Gibson cloning.

For complementation constructs, the recodonized *exp1* gene was PCR amplified using primers to append the Ty sequence and cloned via XhoI and XmaI into p1xNLS-FRB-mCherry containing yDHODH as a resistance marker and different promoters (*nmd3*, *sfa32*, *hsp86*) driving expression of the expression cassette [20]. An AvrII site was (introduced with the primer) before the Ty sequence to allow for further cloning. This resulted in plasmid pEXP1comp. Mutated, chimeric, and deletion constructs were generated using overlapping primers detailed in S1 Table and cloned into pEXP1comp using XhoI and AvrII. mScarlet [42] was synthesized by GenScript and cloned together with SBP1 (using Gibson) into p1xNLS-FRB-mCherry$^{nmd3}$ to generate pSBP1-mScarlet. In this vector, the XhoI and AvrII restriction sites were used to exchanged SBP1 with the first 219 bp of PF13_0191 [43] to obtain the soluble reporter in the PV (SP-mScarlet) or with recodonized *exp1* to obtain the EXP1wt-mScarlet complementation construct. To obtain EXP2-GFP expressed under the *nmd3* promoter, EXP2 was synthesized with a different codon usage (GenScript) and PCR amplified with overhanging sequences to fuse it with GFP and clone it into p1xNLS-FRB-mCherry$^{nmd3}$ digested with XhoI and XmaI using Gibson cloning.

## *P. falciparum* culture and transfection

Blood stages of *P. falciparum* parasites (strain 3D7) were cultured in human RBCs (O+) (transfusion blood, Universität Klinikum Eppendorf, Hamburg) or obtained from the NIH IRB-approved Research Donor Program in Bethesda. Cultures were maintained at 37°C in an atmosphere of 1% $O_2$, 5% $CO_2$, and 94% $N_2$ and cultured using RPMI complete medium (Applichem, Darmstadt, Germany) containing 0.5% Albumax (Invitrogen, Carlsbad, CA) according to standard procedures [44]. For transfection of episomal constructs, Percoll-enriched synchronized mature schizonts were electroporated with 50 μg of plasmid DNA using a Nucleofector II (Lonza) [45]. Selection was done either with 4 nM WR99210 (Jacobus Pharmaceuticals, USA), 2 μg/ml Blasticidin S (Life Technologies, USA), or 0.9 μM DSM1 (BEI Resources; https://www.beiresources.org). For generation of stable integrant cell lines, parasites containing the episomal plasmids selected with WR were grown with 400 μg/ml G418 (Sigma-Aldrich, St. Louis, MO) to select for integrants carrying the desired genomic modification as described previously [20]. For SLI-TGD, a total of 6 independent 2-ml cultures containing the episomal plasmid were selected under G418. To confirm correct integration, genomic DNA from parasites selected under G418 was prepared with a QIAamp DNA Mini Kit and analyzed by PCR using primers specific for the 5' and 3' integration junctions of the *exp1* locus and primers to detect the original locus.

## Generation of condΔEXP1 parasites and DiCre-mediated excision to obtain ΔEXP1 parasites

The parasites containing the integrated pSLI-*exp1-loxP* construct were transfected with pSkip-Flox [20] using 2 μg/ml Blasticidin S to obtain a line expressing the DiCre fragments. To induce excision of the floxed copy of *exp1* in the resulting cell line (condΔEXP1), the parasites were synchronized twice with 5% sorbitol with a time interval of 5 hours, after which the culture was split into 2 dishes of which one dish received rapalog (Clontech, Mountain View, CA) to a final concentration of 250 nM. The untreated dish served as control culture. The rapalog stock (500 mM in DMSO) was stored at −20°C and diluted 1:20 in RPMI as a working solution as described previously [20]. Parasites were cultured in the presence of rapalog for 48 hours. Cultures were synchronized with sorbitol at the beginning of the new cycle to obtain ring stages without EXP1. The parasites of this culture starting the cycle without EXP1 (termed ΔEXP1 parasites) were used for all experiments if not otherwise stated.

For generation of EXP1-Ty-complementation and marker expression cell lines, the cell line condΔEXP1 was transfected with the corresponding yDHODH plasmids (see S1 Table) and selected with 0.9 DSMI μM (BEI Resources).

## Live cell imaging and confocal microscopy

Fluorescence microscopy was done as previously described [46]. Parasites were incubated with 1 μg/ml DAPI in culture medium for 10 minutes to stain nuclei and analyzed using a Zeiss Axioscope M1 equipped with a 100X/1.4 numerical aperture oil immersion objective. A Hamamatsu Orca C4742-95 and the Zeiss Axiovision software were used for collecting images. Images were processed with Corel PHOTO-PAINT X6 (https://www.coreldraw.com).

For counting of nuclei, ΔEXP1 and control parasites were stained with 1 μg/ml DAPI 36 to 44 hours post invasion (h.p.i.) in the second cycle on rapalog (ΔEXP1 parasites), and the number of nuclei was counted by 2 different analysts blinded to the identity of the sample. To quantify the localization of EXP2-GFP after depletion of EXP1, 3 different analysts counted cells (*N* = 25) scoring the number of cells with typical PVM localization (uninterrupted GFP signal

surrounding more than 50% of the parasite) or cells with an aberrant localization (uninterrupted GFP signal surrounding less than 50% of the circumference of the parasite). The distribution of endogenous EXP2 was similarly scored by one analyst using IFA samples. Data were analyzed using Graph Pad Prism 6.07 (Graph Pad Software, https://www.graphpad.com).

For time-lapse imaging, parasites were synchronized using 5% sorbitol, the culture was split into 2 dishes of which one received rapalog to 250 nM and a control without rapalog. After 48 hours, the resulting ΔEXP1 and control parasites were coated onto the bottom of a sterile, uncoated, hydrophobic, high, 35 mm μ-Dish (Ibidi) subdivided in 4 chambers using culture grade 0.5 mg/ml concanavalin A (Sigma) dissolved in dH2O as described previously [46]. Briefly, the concanavalin A (Sigma) was added to the dish surface for 10 minutes at 37˚C, washed off using PBS, and the culture, resuspended in sterile PBS, was added and allowed to settle for 15 minutes, using the different chambers of the dish for ΔEXP1 and control parasites. Nonbound cells were washed off using DPBS, and prewarmed phenol red-free culture medium was added to the dish. Cells were viewed at 37˚C using an Olympus FV1000 confocal microscope equipped with an Olympus Cellcubator. Using the multi-area time-lapse function of the Fluoview software and a motorized stage, at least 10 fields (containing 10–20 infected RBCs) were observed for each condition. Control and ΔEXP1 parasites were imaged simultaneously in the different chambers of the same dish for a period of 76 hours, and images were collected with a 1-hour interval. Focus was maintained using the Olympus ZDC autofocus system. An Olympus 60x/1.35 plan S apo oil immersion lens and Fluoview software version 1.7b was used. Parameters for image collection were usually 4–8 μs laser dwell time, 512 × 512 dpi, 16–32 z-stacks (0.38 μm step size), a zoom level of 3–5, and a 559 nm laser at 1%–5%. The time-lapse experiments were analyzed and processed in Imaris 7.7.2 (Bitplane). Image series were cropped with Image J (https://imagej.nih.gov.ij/), and single images were processed in Corel Photo-Paint X6 (https://www.coreldraw.com).

To analyze the phenotype in ring stages of control and ΔEXP1 parasites, individual ring stages were scored in Imaris at every hour of the time-lapse experiment for ameboid shape, shape change compared to previous time point, position change in the RBC (more than half a cell diameter compared to previous time point), and hugging (defined as close apposition of parasite to RBC periphery), and the frequency was calculated for the total of time intervals examined for a given cell. Time to develop to trophozoite stage from start of the experiment was recorded by determining the time point when parasites contained a clear focus of hemozoin.

### Flow cytometry growth assays and Giemsa stages

For flow cytometry (FC) growth curves, parasitemia was measured by FC and adjusted to 0.1%, and the parasites were divided in two 2-ml dishes (one with 250 nM and a control without rapalog). To follow the growth of the culture using Giemsa smears, the parasitemia was adjusted to 1%, and smears were collected after the intervals indicated. Medium was changed daily, and rapalog was added freshly every day. For the FC curves, the parasitemia was measured as previously described [20]: 20 μl resuspended parasite culture was incubated with dihydroethidium (Cayman Chemical, Ann Arbor, MI) and Hoechst (Cheomdex, Switzerland) at a final concentration of 4.5 μg/ml and 5 μg/ml, respectively, in RPMI for 20 minutes at room temperature protected from light. Before measuring, the cells were fixed with RPMI containing 0.003% glutaraldehyde. For every sample, 100,000 events were recorded using aLSRII flow cytometer (Beckton Dickinson), and parasitemia was determined with the FACS Diva software.

For measuring the capacity to complement ΔEXP1 parasites, the growth of the cell lines with the complementation constructs was assayed over 5 days using the FC growth assay starting before excision of *exp1* (cycle 0). The parasitemia was adjusted to 0.1% and divided in two 2-ml dishes, one with 250 nM rapalog and a control without rapalog. The parasitemia at day 5 was compared to that of the control. At least 4 independent replicas were analyzed for each cell line. To calculate the complementation activity for each construct, the level of growth (parasitemia at day 5 rapalog-treated/growth at day 5 unexcised) was compared to the percentage of complementation of the EXP1wt^mid complementation construct. Data were analyzed with Graph Pad Prism version 6.07 (https://www.graphpad.com) and presented as mean ± SD.

To assess growth in presence of reducing agents, 4-day FC growth assays were performed with matched synchronous ring stages starting with 0.1% parasitemia after one cycle ± rapalog (i.e., using parasites already starting without EXP1 and their matched controls). RPMI was supplemented with Trolox, ascorbic acid, N-acetylcysteine, and cysteine each at a final concentration of 100 μM. Every day, the parasitemia was measured by FC, and fresh medium with supplements was added. For examination of the effect of E64 on growth of ΔEXP1 parasites, ring synchronized parasites at a parasitemia of 1% after one cycle ± rapalog were grown overnight with and without 1 μM E64 (Sigma). The next day, cultures were thoroughly washed to remove E64 and further cultured without the inhibitor. In parallel, the same procedure was carried out with the same cell culture after one cycle ± rapalog but pretreated with 1 μM E64 for 2 hours prior to a pulse of 50 nM DHA (Adipogen, Switzerland). After 3 hours, DHA-treated cultures were washed extensively and further cultured without the drug. The parasitemia was measured each day for 72 hours. The survival rate was calculated as parasitemia of DHA treated or rapalog treated compared to parasitemia of the respective control ± E64.

To evaluate sensitivity to low nutrient conditions, synchronous ring stages after 1 cycle ± rapalog were grown in complete amino acid–restricted medium and complete RPMI medium or medium containing 75 μM NaN₃ (Sigma). To obtain amino acid restricted RPMI medium, complete medium was added in a 1/20 dilution to glucose and amino acid–free RPMI medium 1640 (US Biological). This resulted in a final concentration of 6 mM glucose and 1:20 of the concentration of every amino acid found in standard RPMI complete. Parasitemia was measured after 2 growth cycles using FC (day 5). Relative growth was calculated as parasitemia in restricted medium and NaN₃ containing medium compared to parasitemia of respective control in complete medium without NaN₃.

To evaluate stage distribution of parasites in low amino acid concentration, synchronous ring stages of ΔEXP1 complementation lines were grown for 40 hours with and without rapalog. For this, schizonts that had been grown ± rapalog were Percoll (GE Healthcare, Sweden) purified and allowed to invade while shaking at 37˚C at 750 rpm for 30 minutes in complete and amino acid–restricted medium and further cultured for 3 hours in the respective medium. Rings 0–3 h.p.i. were sorbitol synchronized, and the cultures were continued in the respective medium. Smears were collected after 18, 22, 26, and 30 h.p.i. Stages were counted microscopically, and percentage of ring and trophozoite (parasites containing a clear focus of hemozoin) stages was calculated for every time point.

## Production and purification of antisera

Specific antisera to detect ETRAMP4 and ETRAMP10.1 were raised against the C-terminal domains produced as recombinant GST fusion proteins in *Escherichia coli* as described previously [10]. Rabbit antisera were raised commercially (Eurogentec) and purified over GST-sepharose columns (Genscript) containing the recombinant antigen crosslinked to the column according to established procedures using 30 mM DMP (Thermo Scientific) in 0.2 M

Triethanolamine [47]. Briefly, the crude antiserum was diluted 1/10 in 1x TBS (20 mM TrisHCl [pH 7.0], 150 mM NaCl) containing 1% bovine serum albumin (BSA) and twice passed over the resin containing recombinant crosslinked GST to deplete antibodies binding GST. The flow through was collected and passed through a column containing resin with the corresponding recombinant GST fusion protein crosslinked to it. The antibodies on the resin were washed once with TBS containing 0.1%Triton-X-100, 5 times with TBS, 2 times with 0.1x TBS, and once with 0.1x TBS containing 0.1% Triton-X-100. Thereafter, the bound antibodies were eluted 10 times with 1 ml 0.1 M Glycine (pH 2.5), which was collected in tubes containing 25 μl of 1 M TrisHCl (pH 9.0). Dilutions of eluate 1 and 2 were used for all experiments. Animal handling and immunization at Eurogentec were carried out in accordance with good animal practices according to the Belgian national animal welfare regulations for Eurogentec SA, Seraing and approved by the ethics committee (CE/Sante/E/001) of the Centre d'Economie Rurale (CER Groupe, Marloie, Belgium). At the time of these immunizations, Eurogentec followed the European Union directive 86/609.

### IFAs

IFAs to assess the location of the endogenously HA-tagged EXP1 or the TY-tagged complementation constructs were performed in suspension with Compound 2 [48]-stalled schizonts to differentiate protein located at the PPM from that located at the PVM. For this, trophozoite stages were treated with Compound 2 (1 μM) overnight, and arrested schizonts were harvested, washed in PBS, and fixed with 4% paraformaldehyde/0.0075% glutaraldehyde in PBS [49]. Cells were permeabilized with 0.1% Triton X-100 in PBS, blocked with 3% BSA in PBS, and incubated for 1 hour with primary antibodies: rat α-HA (Roche, Mannheim, Germany) (1:500), rabbit α-HA (Cell Signaling, USA) (1:500), mouse α-Ty (Sigma) (1:20,000), human α-MSP1 (PPM marker [1:1,000]) [50] diluted in 3% BSA in PBS. Cells were washed 3 times with PBS and incubated for 1 hour with Alexa 488 nm or Alexa 594 nm conjugated secondary antibodies specific for human, mouse, rabbit, or rat IgG (Invitrogen) diluted 1:2,000 in 3% BSA in PBS and containing 1 μg/ml DAPI. Cells were directly imaged after washing 5 times with PBS.

For IFAs detecting PVM markers and exported proteins, the condΔEXP1, condΔEXP1+-EXP2-GFP, condΔEXP2, and condΔEXP2+EXP1-Ty cell lines were grown for 48 hours on rapalog (250 nM) to obtain the corresponding ΔEXP1 or ΔEXP2 parasites. Rings in the second cycle were directly used or parasites were synchronized and allowed to develop to trophozoite stages. Cells were fixed and permeabilized as described above and incubated with rabbit α-SBP1 (C) (1:2,000) [31], rabbit α-KAHRP (1:500) (a kind gift of Prof. Brian Cooke), rabbit α-REX1 (1:10,000) [31], mouse α-REX2 (1:500) [51], mouse α-MSRP6 1:250 [43], mouse α-ETRAMP5 (1:500) [52], rabbit α-ETRAMP4 (1:500), rabbit α-ETRAMP10.1 (1:500), mouse α-GFP (Roche) (1:500), rabbit α-GFP (Thermo Fischer, USA) (1:500), mouse α-Ty (Sigma) (1:20,000), and mouse monoclonal 7.7 α-EXP2 (1:2,000).

Cultures containing gametocytes were fixed in suspension as described above, air-dried as thin films on 10-well slides (Thermo Fischer), and fixed in 100% acetone for 30 minutes at room temperature. IFAs were labelled with mouse α-Pfs16 1:1,000 [53], rat α-Pfg377 1:1,000 [54], and rabbit α-spectrin 1:500 (Sigma).

### Staining of parasite membranes using lipid dyes

Bodipy-TR-C5-ceramide (Invitrogen) staining was performed using a concentration of 2.5 μM (stock 5 μM) in RPMI as previously described [46] in ring and trophozoites of condΔEXP1 parasites and condΔEXP1 expressing EXP2-GFP after 1 cycle ± rapalog. For Lyso PC labelling, TopFluor LysoPC (Avanti Polar Lipids, Alabaster, AL) (1 mM stock in methanol) was

resuspended in PBS to a final concentration of 20 μM, added to ring and trophozoite stages of control and ΔEXP1 parasites, and incubated for 15 minutes at 37˚C. All microscopy images of the lipid dye stained parasites were recorded with the same acquisition settings and exposure time. Number of protrusions in each parasite were counted, and data were analyzed with Graph Pad Prism 6.07 (Graph Pad Software, http://www.graphpad.com).

### Electron microscopy

Control and ΔEXP1 parasites were harvested 14 to 24 h.p.i. Cells were fixed with 2.5% glutaraldehyde (Electron Microscopy Sciences, USA) in 50 mM cacodylate buffer (pH 7.4) for 1 hour at room temperature. Cells were post fixed with 2% $OsO_4$ in $H_2O$ (Electron Microscopy Sciences) for 40 minutes at 4˚C in the dark, contrasted with 0.5% uranylacetate (Electron Microscopy Sciences) for 30 minutes at room temperature, and dehydrated through increasing concentrations of ethanol. Following embedding in epoxy resin (EPON) (Roth, Karlsruhe, Germany), 60 nm sections were generated with an Ultracut UC7 (Leica) and examined with a Tecnai Spirit transmission electron microscope (FEI), equipped with a LaB6 filament and operated at an acceleration voltage of 80 kV.

### Solubility assays of EXP1 constructs

For saponin lysis to separate PV proteins from membrane-associated proteins, Percoll-enriched trophozoites (from 5–10 ml culture with a parasitemia of 5%–10%) of the cell lines expressing complementation Ty constructs were washed twice with PBS and incubated on ice for 10 minutes with 100 μl PBS containing a final concentration of 0.015% saponin (Sigma, Steinheim), followed by centrifugation at 16,000$g$ for 5 minutes. The supernatant (containing PV and host cell soluble proteins) was transferred to a new tube and mixed with protease cocktail inhibitor (Roche) and 1 mM PMSF and reducing sodium dodecyl sulfate (SDS) sample buffer. The parasite pellet (containing membrane proteins and parasite proteins confined within the PPM) was washed once with DPBS and then resuspended in 100 μl of protein lysis buffer (0.5x PBS/4% SDS/0.5% Triton X-100) containing complete protease inhibitor cocktail. The pellet lysate was cleared using a centrifugation at 16,000$g$ for 5 minutes, and the supernatant was transferred to a second tube, and reducing SDS sample buffer was added. Equivalent volumes were analyzed by SDS-polyacrylamide gel electrophoresis (PAGE) and western blotting.

To assess leakage of PV proteins in the SP-mScarlet expressing ΔEXP1 parasites, the host cell cytosol and the parasite including the PV content was first separated by tetanolysin lysis as follows: Percoll-enriched trophozoites were generated from 5–10 ml of parasite culture (5%–10% parasitemia), washed with PBS, and incubated in 100 μl PBS containing 1 HU tetanolysin (Santa Cruz Biotechnology, USA) for 5 minutes at 37˚C. The supernatant (containing soluble proteins from the host cell) was transferred to a new tube and mixed with protease cocktail inhibitor (Roche) and PMSF 1 mM and reducing SDS sample buffer. The parasite pellet was processed as described above for saponin lysis.

For total parasites extracts, parasites were released from RBCs by incubation in 0.03% saponin in PBS for 10 minutes on ice followed by 3 washes with PBS. Proteins were then extracted with protein lysis buffer in the presence of protease cocktail inhibitor (Roche) and 1 mM PMSF. After centrifugation at 16,000$g$ for 5 minutes, reducing SDS sample buffer was added to the supernatant, and the sample was analyzed by SDS-PAGE and immunoblotting.

To test the membrane extractability of EXP2 after removal of EXP1, control and ΔEXP1 trophozoites were Percoll purified from 5–10 ml of parasite culture (5%–10% parasitemia), washed with PBS, and lysed in 100 μl 5 mM Tris-HCl (pH 8.0)/1 mM EDTA containing

protease inhibitor cocktail (Roche) for 10 minutes on ice. Lysates were frozen at −80°C, thawed, and centrifuged 5 minutes at 16,000$g$. The resulting pellet was washed once with 5 mM Tris-HCl (pH 8.0)/1 mM EDTA and resuspended in 200 μl 5 mM Tris-HCl (pH 8.0)/1 mM EDTA. The suspension was divided into 4 tubes (50 μl each) and centrifuged 5 minutes at 16,000$g$. The 4 pellets were resuspended in one each of the following solutions: (1) 0.5 x PBS/4% SDS/0.5% Triton X-100 containing protease inhibitor cocktail (Roche) (corresponding to the total control); (2) 0.1 M Na$_2$CO$_3$ (pH 11.5); (3) 8 M urea/5 mM Tris-HCl (pH 8.0)/1 mM EDTA; and (4) Triton 1% in 1x PBS containing protease inhibitor cocktail (Roche). The samples were incubated on ice for 30 minutes, except for the first (total control), which was directly frozen. Tubes 2, 3, and 4 were centrifuged 5 minutes at 16,000$g$ and the supernatant (extracted proteins) transferred into a fresh tube. The corresponding pellets were resuspended in 0.5x PBS/4% SDS/0.5% Triton X-100 containing protease inhibitor cocktail. Equivalent volumes of supernatant and pellet (or supernatant only for total control) were analyzed by SDS-PAGE and western blotting. For densitometric analyses, the intensity of EXP2 signal in supernatant and pellet was measured. A ratio supernatant/pellet EXP2 in every fraction was calculated and normalized to the ratio of the BIP signal.

## Formaldehyde in vivo cross linking

In vivo cross linking was performed as described previously [12]. Parasite cultures (10 ml, 3%–5% parasitemia) were washed twice with PBS and split into 2 tubes. The cells were resuspended in PBS, and to one tube, formaldehyde (PFA) was added to a final concentration of 1%. The samples were incubated at 37°C for 30 minutes, and then Tris-HCl (pH 8.0) was added to 30 mM to quench the reaction. Both samples were centrifuged at 3,000$g$ for 5 minutes followed by lysis in 10 ml of 10 mM Tris-HCl (pH 8.0) on ice for 1 hour. The sample was centrifuged at 5,000$g$ for 15 minutes, and the pellet was washed 3 times in 1.5 ml ice-cold PBS with centrifugations at 16,000$g$. The layer on top of the pellet representing erythrocyte ghost membranes was removed, and the final pellet was resuspended in 2 volumes of protein lysis buffer and stored at −80°C. Equivalent volumes of cross-linked and non–cross-linked samples were analyzed by immunoblotting.

## Immunoblotting analyses

Protein samples were resolved by SDS-PAGE and transferred to Amersham Protran membranes (GE Healthcare, Germany) in a tankblot device (Bio-Rad) using transfer buffer (0.192 M Glycine, 0.1% SDS, 25 mM Tris) with 20% methanol or 10 mM CAPS buffer (pH 11) without methanol. Membranes were blocked, and antibodies were diluted in PBS containing 5% skim milk. Washing steps were done with PBS. Primary antibodies were applied in the following dilutions: mouse α-Ty (Sigma), 1:20,000; rat α-HA (Roche), 1:1,000; rabbit α-HA (Cell Signaling), 1:1,000; mouse α-GFP (Roche), 1:1000; rabbit α-GFP (Thermo Fischer), 1:2,000; rat α-RFP (Chromotek, Germany), 1:1,000; rabbit α-SERA5, 1:2,000 [31] rabbit α-REX3, 1:2,000 [51]; rabbit α-SBP1(C),1:2,000 [31]; rabbit α-aldolase, 1:2,000 [31]; rabbit α-ETRAMP4, 1:500; and rabbit anti-BIP, 1:2,000 [55]. After 3 washes with PBS, horseradish peroxidase-conjugated secondary antibodies goat α-rat (Dianova, Hamburg, Germany) and goat α-mouse (Dianova; 1:3,000) and donkey α-rabbit (Dianova; 1:2,500) were incubated for 2 hours to overnight. Detection was done using enhanced chemiluminescence (Bio-Rad/Thermo Fischer), and signals were recorded with a ChemiDoc XRS imaging system (Bio-Rad). Densitometric analyses were performed with Image Lab software 5.2 (Bio-Rad). Intensity of Ty signal of EXP1wt-Ty constructs expressed under the different promoters was normalized to the BIP signal, and the ratio was compared to that of the EXP1wt-Ty$^{mid}$, which was set to 100%.

## Quantification of ROS

Control and ΔEXP1 ring parasites were cultured overnight in the presence of 200 μM 5-ALA (Sigma) and further cultured in the presence of rapalog. The next day, after 2 washes with DPBS, the resulting trophozoite-stage parasites were incubated for 30 minutes with 5 μM $CM_2$-DCFDA (Invitrogen) in DPBS at 37˚C protected from light. Cells were washed twice with DPBS and further cultured in RPMI for 2 hours at 37˚C under standard conditions. Parasites with similar size were imaged, and fluorescence was captured with the same acquisition settings to obtain comparable measurements of the fluorescence intensity. Fluorescence intensity (integrated density) was measured with Image J [56], and background was subtracted in each image. The data were analyzed with Graph Pad Prism version 6.07 (http://www.graphpad.com).

To quantify the number of parasites exposed to oxidative stress, the parasites were incubated with $CM_2$-DCFDA and further cultured in RPMI with no supplements or in the presence of diamide (100 μM) (Sigma) or Trolox (6-hydroxy-2,5,7,8-tetramethylchroman-2-carboxylic acid) (100 μM) (Sigma). Before analysis, the parasites were stained with 5 μg/ml Hoechst in RPMI for 20 minutes in the dark, and the number of cells that were $CM_2$-DCDFA positive and 5-ALA positive was quantified by FC with an LSRII flow cytometer (BD Biosciences, Franklin Lakes, NJ). The percentage of cells exposed to oxidative stress was calculated as number of $CM_2$-DCDFA and 5-ALA positive cells compared to the number of all 5-ALA positive cells. Mean fluorescence of $CM_2$-DCDFA in the 5-ALA positive cells was estimated with Flow Jo 10 (https://www.flowjo.com), and data were analyzed with Graph Pad Prism 6.07 (http://www.graphpad.com).

## 5-ALA uptake assay

Synchronized ring stages after one cycle ± rapalog were cultured overnight in presence of 200 μM 5-ALA (Sigma). Parasites were stained with 1 μg/ml DAPI, and trophozoite stages were imaged using the same acquisition settings of the fluorescence microscope. Parasites were stained with Hoechst, and the PPIX-positive cells were detected with an LSRII flow cytometer (BD Biosciences) detecting PPIX emission with the 532 nm laser through a 605/40 nm bandpass filter and Hoechst emission with the 406 nm laser through a 440/40 nm bandpass filter to quantify the number of PPIX-positive and Hoechst-positive cells. Erythrocyte doublets were excluded using an FCS-A versus FCS-H display. Data were analyzed by BD FACS Diva software (BD Biosciences), FlowJo 10 (https://www.flowjo.com), and Graph Pad Prism 6v.07 (Graph Pad Software, http://www.graphpad.com).

## Electrophysiology

To initiate the KO of condΔEXP1+SP-mScarlet parasites, late-stage–infected RBCs (65% Percoll interface) were left to infect new RBCs overnight. New rings were isolated (pellet of 65% Percoll), and 250 nM rapalog were added (day 1) to one culture dish; a second dish served as control without rapalog. On day 3, parasites were patch clamped. The control was cultured in parallel without the addition of rapalog.

The PVM nutrient-permeable channel was detected in the PVM of parasites released from the host RBC after Percoll isolation and incubation in an isotonic high potassium buffer (140 mM KCl, 5 mM NaCl, 0.4 mM CaCl2, 0.4 mM MgCl2, 25 mM HEPES, 4.5 mg/ml glucose, 0.5% Albumax II, 66 nM phalloidin-Alexa488 [Invitrogen]) [57]. The phalloidin was added to visualize the attached host RBC and confirm release of the parasite. The parasites were transferred on the microscope in 150 mM NaCl, 5 mM KCl, 1.4 mM CaCl2, 1 mM MgCl2, 20 mM HEPES NaOH (pH 7.4), and 4.5 mg/ml glucose. The patch pipette (borosilicate glass) was

pulled with a Model P80 (Sutter instrument) to 15–20 MΩ and filled with 155 mM CsCl, 1.4 mM CaCl2, 1 mM MgCl2, and 20 mM HEPES NaOH (pH 7.4). Electrophysiology data were recorded using an Axopatch 200B amplifier equipped with a CV203BU head stage (Molecular Devices, San Jose, CA). The signal was filtered at 10 kHz (8-pole Bessel) and digitized at 50 kHz using a Digidata 1550B (Molecular Devices). Error bars in the detection frequency bar graph were calculated in Excel (Microsoft); the *P* value was calculated in R (version 3.5.0, R Core Team) using the Fisher test function.

## Co-IP assays

The cell line condΔEXP1 expressing EXP2-GFP was sorbitol synchronized, and ring-stage parasites were adjusted to approximately 5% parasitemia. The next day, late trophozoites were harvested and washed twice with DPBS. The culture was cross-linked with 0.5 mM dithiobis (succinimidylpropionate) (DSP, from a 20 mM stock in DMSO) (Pierce, USA) in DPBS for 30 minutes at room temperature, and the reaction was quenched with PBS containing 25 mM Tris-HCl. Cross-linked infected RBCs were purified in a Percoll gradient, washed twice with DPBS, and lysed with RIPA buffer (10 mM Tris HCl [pH 7.5], 150 mM NaCl, 0.1% SDS, 1% Triton) containing protease inhibitor cocktail (Roche) and 1 mM PMSF. After 2 freeze-thaw cycles, lysates were cleared by centrifugation at 16,000*g* for 10 minutes, and the supernatant was diluted 1:2 with RIPA buffer without detergents. The supernatants were incubated with 25 µl of mouse monoclonal anti-HA beads (Pierce, USA) or anti-GFP beads (Chromotek, Germany) for 3 hours at 4˚C. Samples of input and post binding extracts were saved for immunoblot analysis. Beads were recovered by centrifugation and washed 5 times with RIPA buffer. Proteins were eluted in 50 µl 4x SDS sample buffer at 85˚C for 5 minutes. Equal volumes of input post binding extract and bound fractions were subjected to western blot analysis.

## RSAs and determination of DHA IC50

RSAs were performed according to established procedures [30]. Briefly, synchronous ring stages of ΔEXP1 complementation lines were grown for 40 hours with and without rapalog. Kelch13[C580Y] [20] was analyzed as positive control for DHA resistance. Percoll-purified schizonts ± rapalog were allowed to invade fresh RBCs shaking at 37˚C at 750 rpm for 30 minutes and further cultured for 3 hours. Rings 0 to 3 h.p.i. were obtained by sorbitol treatment. These rings were exposed to 350 nM DHA (Adipogen, Switzerland) for 6 hours alongside an untreated control. Following removal of DHA by thorough washing, parasites were cultured for 66 hours under standard conditions. The number of viable parasites was counted in 10,000 erythrocytes in Giemsa smears to calculate survival rate as parasitemia of ± rapalog DHA-treated compared to parasitemia of ± rapalog DHA-untreated cultures.

For determination of IC50, the different ΔEXP1 complementation parasites were grown with and without rapalog for 48 hours. Ring stages were sorbitol synchronized, adjusted to a start parasitemia of approximately 1%, and cultured with increasing concentrations (0 to 50 nM) of DHA. The medium was changed after 24 hours, and fresh DHA was added. The parasitemia was measured by FC as described above after 48 hours, and the IC50 was calculated using GraphPadPrism version 6.07.

## Gametocyte induction in ΔEXP1 parasites

ΔEXP1 and control parasites were sorbitol synchronized and grown in RPMI supplemented with 50 mM N-acetyl glucosamine (Serva) for 5 days without diluting the culture. Samples were collected first at day 3 for Giemsa smears and fixed for IFA in suspension or dried, and acetone-fixed for detection of early gametocytes with α-Pfs16. N-Ac-Gluc was removed after 5

days, and the parasites were further cultured. On day 8 after addition of N-Ac-Gluc, samples were collected for IFA labelled with Pfg377 to detect late gametocytes. RBC spectrin was labelled to count the number of gametocytes per 1,000 RBCs. Percentage of Pfs16- and Pfg377-positive cells in the rapalog-treated culture was compared to that in control parasites to calculate fold reduction of cells positive with the respective antigen. Data were analyzed by Graph Pad Prism version 6.07 (http://www.graphpad.com).

## Supporting information

**S1 Fig. Conditional EXP1 KO.** (A) Schematic representation of the SLI strategy to obtain a cell line for DiCre-based conditional KO of *exp1*. Top shows endogenous locus and plasmid pSLIΔEXP1cond. Light yellow box: cell line after SLI. Light orange box: the cell line transfected with pSkipFlox [20] and after induction of DiCre. "loxP" indicates the loxP site; asterisks indicates in-frame stop codon. Cre 60–343, Cre 19–59: Cre fragments; arrows, primers P1, P2, P3, and P4. (B) PCR on gDNA of condΔEXP1 and 3D7 parasites using the primers indicated in (a) confirming: 5'Int, 5'integration; absence of 'original locus'; 3'Int, 3' integration. (C) FC growth curves of synchronous condΔEXP1 ring stage parasites grown ± rapalog over 5 days (addition of rapalog starting day 1). Blue arrow indicates start of cycle without EXP1 ('ΔEXP1 parasites'). One representative of $n$ = 3 experiments is shown. (D) IFA images of condΔEXP1 trophozoites grown for 24 hours (first cycle) or 72 hours (second cycle) with and without rapalog (control) probed with α-HA to detect EXP1*-HA and α-myc for the truncated EXP1 stub in the control. Note that after excision (rapalog), the stub contains both, myc- and HA-tag, and it will be recognized by both antibodies (see panel A). Nuclei were stained with DAPI; scale bars: 5 μm. (E) Number of nuclei in DAPI-stained control and ΔEXP1 parasites (rapalog) 40 h.p.i. One representative of $n$ = 3 independent biological replicas. (F) Long-term FC growth curve of synchronous ring control and ΔEXP1 parasites (rapalog) after depletion of EXP1 (blue arrow) at the times indicated. Blue box shows zoom of restricted to 20% parasitemia on the y-axis to show raise in the control in early time points. Orange box, PCR with primers P1 and P2 (see panel A) from gDNA of control and rapalog-treated ΔEXP1 parasites on day 9. Mean of $n$ = 2 independent experiments. Error bars indicate SD. (G) Transmission electron microscopy images of control and ΔEXP1 parasites (rapalog) 18 to 24 h.p.i. showing hugging in ΔEXP1 parasites. Scale bar, 500 nm. (H) Left, live cell images of control and ΔEXP1 (rapalog) parasites labelled with TopFluor Lyso PC (Lyso PC). Yellow arrows, tubo-vesicular network. Graph: quantification of protrusions per cell in $n$ = 22 control cells and $n$ = 38 ΔEXP1 parasites from 2 independent experiments. (I) Upper panel, live cell images of control and ΔEXP1 parasites (rapalog) expressing SPmScarlet. DAPI, nuclei. Light blue arrow shows a bleb. Lower panel: immunoblot of protein extracts from RBCs infected with these parasites, permeabilized with tetanolysin and separated into SN (host cell cytosol) and P, pellet (parasite within PVM). α-REX3, control for host cell cytosol; α-BIP, loading control. (E, H) green lines indicate mean and error bars SD; two-tailed unpaired $t$ test, $P$ values are indicated. BSD, blasticidine deaminase; DIC, differential interference contrast; EXP1*, recodonized *exp1*; HA, triple hemagglutinin; hDHFR: human dihydrofolate reductase; L, linker; Neo: neomycin phosphotransferase; NLS: nuclear localization signal; asterisk, in frame stop codon; SP, signal peptide; T2A, skip peptide; TM, transmembrane domain.
(PDF)

**S2 Fig. Genetic complementation of ΔEXP1 parasites.** (A) Schematics of the complementation constructs expressed in condΔEXP1 parasites. Numbers refer to amino acids of the domains shown in the legend. (B) Relative activity of the complementation constructs. Except where otherwise indicated, constructs were expressed under the *nmd3* (mid) promoter. Each

data point (red dot) shows growth of rapalog-treated versus unexcised parasites at the end of a 5-day growth assay relative to the growth of the wt construct. Green lines indicate activity of EXP1wt$^{nmd3}$ (mid) (set as 100%) and absence of activity (ΔEXP1) set as 0%; $n \geq 4$ independent experiments per cell line. Error bars indicate SD. (C) Mean ± SD of relative growth versus unexcised (control) and mean of relative complementation versus EXP1wt$^{mid}$ (used in panel B and in the graphs in Figs 2 and 3); $n$ numbers evident in (B). (D) Percentage of rings and trophozoites of tightly synchronous parasites of ΔEXP1 and complemented ΔEXP1 parasites at the time points indicated after invasion (after an initial cycle ± rapalog). Mean of $n = 2$ independent experiments. (E) Amino acid sequence of the central region (including the TM domain) of EXP1 and selected ETRAMPs from *P. falciparum*. Boxes show conserved G, S, and T rich regions. Hydrophobic residues, red; positively charged, pink; negatively charged, blue; polar (N, T, G, S, Q, H, and Y), green. (F) Left, alignment of the EXP1 TM region from different *Plasmodium* species. Asterisk, conserved and double dot, partially conserved residues; mutated G, yellow boxes; predicted TM in *P. falciparum* EXP1 is boxed. Right, helical wheel diagram of the *Pf*EXP1 TM domain (numbered from 1 to 23).
(PDF)

**S3 Fig. Localization and solubility of EXP1 complementation constructs.** (A) IFA images of compound 2-arrested condΔEXP1 schizont stages expressing the complementation constructs indicated above each panel (α-HA detects EXP1*-HA; α-Ty1, complementing EXP1 copy; anti-RFP, EXP1mScarlet. α-MSP1, PPM). DAPI, nuclei. Scale bars: 5 μm. (B) Immunoblots of extracts of the cell lines shown in (a). Saponin was used to separate the parasite pellet (P) from the supernatant (SN) containing PV and host cell content. α-Ty1 detects the complementation constructs, anti-RFP, EXP1mScarlet and α-SBP1 was used to detect a membrane-associated control protein. DIC, differential interference contrast.
(PDF)

**S4 Fig. Oxidative stress and protein export in ΔEXP1 parasites.** (A) FC analysis of matching 5-ALA-treated control and ΔEXP1 parasites (rapalog) after incubation with CM-H$_2$DCFDA in RPMI alone or supplemented with diamide or Trolox at the time points indicated. The percentage of cells with oxidative stress corresponds to the number of CM-H$_2$DCFDA positive cells of the total number of 5-ALA-positive cells. Error bars, SD. $n = 4$ independent biological replicas. (B) Fluorescence of control and ΔEXP1 parasites analyzed in (a). Green line, mean; error bars, SD. $n = 4$ independent biological replicas. (C) IFA images of control and ΔEXP1 parasites (rapalog) probed with α-HA (EXP1*-HA), α-SBP1, α-REX1, α-REX2, and α-MSRP6. Size bars, 5 μm. h.p.i., hours post invasion.
(PDF)

**S5 Fig. Localization of EXP2 in ΔEXP1 ring stages.** (A) Live cell images of control and ΔEXP1 (rapalog) ring stages episomally expressing EXP2-GFP$^{nmd3}$. (B) IFA images of control and ΔEXP1 ring stages (rapalog); α-HA detects EXP1*-HA, α-EXP2 detects endogenous EXP2. Nuclei were stained with DAPI. Scale bars: 5 μm. DIC, differential interference contrast.
(PDF)

**S6 Fig. SLI-TGD of ETRAMP5 has no effect on parasite growth.** (A) Schematic representation of SLI-TGD to disrupt *etramp5*. Features as in S1A Fig. (B) PCR on gDNA of ETR5-TGD and wild-type 3D7 parasites confirming: 5'Int, 5'integration; absence of 'original locus'; 3'Int, 3' integration. (C) Live cell images of young trophozoite and schizont stages of ETR5-TGD parasites (fluorescence shows the truncated GFP-tagged protein). DAPI, nuclei; scale bars: 5 μm. (D) Immunoblot of extracts of ETR5-TGD parasites separated into saponin supernatant

(SN, containing PV and host cell content) and parasite pellet (P). α-GFP, detects truncated ETR5; α-BIP: control for the parasite pellet. Asterisk, protein degraded down to GFP; double asterisk, unskipped protein (first T2A, no unskipped product detected at the GFP-Neomycin junction). The truncated protein has no TM and is therefore found in the SN. (E) Left: FC 5-day growth curves of synchronous 3D7 and ETR5-TGD parasites. Mean of $n = 3$ independent biological replicas. Right: fold increase in parasitemia over 5 days for 3D7 and ETR5-TGD parasites measured by FC. Green line indicates mean and error bars SD, two-tailed unpaired $t$ test; $P$ value indicated. DIC, differential interference contrast; ns, not significant.
(PDF)

**S7 Fig. Conditional deletion of *exp*2.** (A) Schematic representation of the SLI strategy to obtain a cell line for DiCre-based conditional KO of *exp2*. Features as in S1A Fig. (B) PCR products from gDNA of condΔEXP2 and wild-type 3D7 parasites confirming: 5'Int, 5'integration; absence of 'original locus'; 3'Int, 3' integration. (C) Strategy to deplete EXP2 from the PVM using synchronized condΔEXP2 ring stages divided into a culture with and one without rapalog. Top: schematic: green boxes and blue line around the parasite signify PVM with EXP2. Mid: PCR with primers P1 and P2 from gDNA 24 hours and 48 hours after addition of rapalog. Original: PCR product for locus with intact *exp2*; excised: PCR product after excision of *exp2*. Bottom: western blot using α-HA to detect EXP2*-HA and α-BIP as loading control. Note that the small truncated fragment after excision of the functional copy becomes HA-tagged (see panel A) but is not detected (likely due to its small size and its instability leading to low abundance, see panel F). (D) Giemsa smears of synchronous ΔEXP2 parasites (rapalog) compared to the controls. Blue arrow indicates start of a new cycle without EXP2. (E) FC growth curves of synchronous ring stage condΔEXP2 parasites grown ± rapalog over 5 days. Blue arrow indicates start of cycle without EXP2. One representative of $n = 3$ experiments. (H) IFA images of control and ΔEXP2 parasites (rapalog) probed with α-HA, which detects full functional (control) or truncated inactivated (rapalog) EXP2-HA and SBP1 (α-SBP1) or REX1 (α-REX1). DAPI, nuclei. Scale bars: 5 μm. Note that the truncated inactive version of EXP2 is not well detected, likely because it is degraded. DIC, differential interference contrast.
(PDF)

**S8 Fig. EXP1–EXP2 interaction analysis and analysis of 5-ALA–treated ΔEXP1 parasites.** (A) Western blot of reciprocal co-IP experiment using α-GFP with extracts of the cell line condΔEXP1+EXP2-GFP$^{nmd3}$ to pull down EXP2-GFP. α-HA detects EXP1*-HA (monomer: asterisk, dimer: double asterisk); α-SERP, soluble PV protein; α-aldolase, cytosolic parasite protein. Input (I): total lysate before IP; post IP (P): lysate after IP; Eluate (E). One representative of $n = 3$ independent experiments. (B) IFA images of EXP1-3xHA$^{endo}$ and EXP2-3xHA$^{endo}$ merozoites probed with α-HA and α-MSP1 (plasma membrane marker). Nuclei were stained with DAPI; scale bar 2 μm. (C) Immunoblot of protein extracts derived from ΔEXP1 (rapa) and control trophozoites. Saponin was used to separate parasite pellet (P) from the supernatant (SN) containing PV and host cell soluble proteins. α-EXP2 detects endogenous EXP2; α-SERP, a soluble PV soluble to control for proper PVM permeabilization; α-BIP, as loading control and α-HA (detecting EXP1*-HA) to show loss of EXP1. One representative of $n = 2$ experiments. (D) Left: immunoblot of protein extracts from ΔEXP1 (rapa) and control trophozoites fractionated in SN and P after hypotonic lysis and extraction with $Na_2CO_3$, urea (peripheral membrane proteins) and Triton x-100 (TX-100, integral membrane proteins). α-EXP2 detects endogenous EXP2 and α-BIP, a parasite-internal peripheral membrane protein. Right: densitometric analysis of EXP2 intensity in SN and P. The ratio SN/ P of the EXP2 signal was calculated for $Na_2CO_3$ and urea and normalized to the ratio of BIP. Green line: mean

of $n$ = 4 independent experiments; error bars, SD. $P$ values were calculated with a two-tailed unpaired $t$ test. (E) FC analysis of 5-ALA–treated control and ΔEXP1 parasites (rapalog) shows that the number of PSAC-positive cells correlates with the number of cells that reached the trophozoite stage, irrespective of whether parasites contained EXP1 or not. Left, gating for Hoechst/PPIX-positive cells (upper right quadrant). Right: quantification of PPIX-positive cells and percentage of trophozoites in the same cultures. Mean of $n$ = 3 independent biological replicates; two-tailed unpaired $t$ test; $P$ values indicated. (F) Gating strategy for quantification of *P. falciparum*–infected RBC cells by FC. Left panel: forward versus side scatter (FSC versus SSC) gating to define population of RBCs and exclude debris. Mid panel, forward scatter height (FSC-H) versus forward scatter area (FSC-A) density plot to define single RBCs and exclude doublets. Right panelL DAPI (Hoechst) versus PE-A (dihydroethidium, DHE) density plot to distinguish infected RBCs from uninfected RBCs. DIC, differential interference contrast.
(PDF)

**S1 Table. Oligonucleotides used in this study.**
(PDF)

**S1 Data. Excel file containing seperate sheets of the numerical data underlying the graphs of the main figures.**
(XLSX)

**S2 Data. Excel file containing seperate sheets of the numerical data underlying the graphs of the supporting information figures.**
(XLSX)

**S1 Raw images. Minimally cropped blots and gels shown in the main figures and supporting information figures.**
(PDF)

## Acknowledgments

We are grateful to Marcel Deponte for critical reading of the manuscript and interpretation of oxidative stress data, to Ralf Krumkamp for assistance with statistical analysis, to Arlett Heiber for purification of ETRAMP antisera, and to Svetlana Glushakova for helpful discussions. We thank Pietro Alano for α-Pfs16 and α-Pfg377 antibodies, Michael Blackman for Compound 2 and α-MSP1 antibodies, Tim Gilberger for α-BIP antibodies, Brian Cooke for α-KAHRP antibodies, and Matthias Marti for Topfluor Lyso PC. Monoclonal antibody 7.7 (α-EXP2) was obtained from The European Malaria Reagent Repository (http://www.malariaresearch.eu). We thank Jacobus Pharmaceuticals for WR99210. DSM1 (MRA-1161) was obtained from MR4/BEI Resources, NIAID, NIH.

## Author Contributions

**Conceptualization:** Paolo Mesén-Ramírez, Tobias Spielmann.

**Data curation:** Bärbel Bergmann.

**Formal analysis:** Paolo Mesén-Ramírez, Thuy Tuyen Tran, Joshua Zimmerberg, Tobias Spielmann.

**Funding acquisition:** Tobias Spielmann.

**Investigation:** Paolo Mesén-Ramírez, Bärbel Bergmann, Thuy Tuyen Tran, Matthias Garten, Jan Stäcker, Isabel Naranjo-Prado, Katharina Höhn, Tobias Spielmann.

**Methodology:** Paolo Mesén-Ramírez, Bärbel Bergmann, Thuy Tuyen Tran, Matthias Garten, Jan Stäcker, Isabel Naranjo-Prado, Katharina Höhn, Joshua Zimmerberg, Tobias Spielmann.

**Project administration:** Paolo Mesén-Ramírez, Joshua Zimmerberg, Tobias Spielmann.

**Resources:** Tobias Spielmann.

**Supervision:** Paolo Mesén-Ramírez, Katharina Höhn, Joshua Zimmerberg, Tobias Spielmann.

**Validation:** Paolo Mesén-Ramírez, Bärbel Bergmann, Thuy Tuyen Tran, Tobias Spielmann.

**Writing – original draft:** Paolo Mesén-Ramírez, Matthias Garten, Joshua Zimmerberg, Tobias Spielmann.

**Writing – review & editing:** Paolo Mesén-Ramírez, Matthias Garten, Jan Stäcker, Isabel Naranjo-Prado, Katharina Höhn, Joshua Zimmerberg.

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
