## [Editor Report · Decision Letter 0]

17 Jun 2019

Dear Dr Spielmann, 

Thank you for submitting your manuscript entitled "EXP1 is critical for nutrient uptake across the parasitophorous vacuole membrane of malaria parasites" for consideration as a Research Article by PLOS Biology.

Your manuscript has now been evaluated by the PLOS Biology editorial staff as well as by an academic editor with relevant expertise and I am writing to let you know that we would like to send your submission out for external peer review.

**Important**: Please also see below for further information regarding completing the MDAR reporting checklist. The checklist can be accessed here: https://plos.io/MDARChecklist

Please re-submit your manuscript and the checklist, within two working days, i.e. by Jun 19 2019 11:59PM.

Kind regards,

Lauren A Richardson, Ph.D,

Senior Editor

PLOS Biology

INFORMATION REGARDING THE REPORTING CHECKLIST:

PLOS Biology is pleased to support the "minimum reporting standards in the life sciences" initiative (https://osf.io/preprints/metaarxiv/9sm4x/). This effort brings together a number of leading journals and reproducibility experts to develop minimum expectations for reporting information about Materials (including data and code), Design, Analysis and Reporting (MDAR) in published papers. We believe broad alignment on these standards will be to the benefit of authors, reviewers, journals and the wider research community and will help drive better practise in publishing reproducible research. 

We are therefore participating in a community pilot involving a small number of life science journals to test the MDAR checklist. The checklist is intended to help authors, reviewers and editors adopt and implement the minimum reporting framework. 

IMPORTANT: We have chosen your manuscript to participate in this trial. The relevant documents can be located here:

MDAR reporting checklist (to be filled in by you): https://plos.io/MDARChecklist

**We strongly encourage you to complete the MDAR reporting checklist and return it to us with your full submission, as described above. We would also be very grateful if you could complete this author survey:

https://forms.gle/seEgCrDtM6GLKFGQA

Additional background information:

Interpreting the MDAR Framework: https://plos.io/MDARFramework

Please note that your completed checklist and survey will be shared with the minimum reporting standards working group. However, the working group will not be provided with access to the manuscript or any other confidential information including author identities, manuscript titles or abstracts. Feedback from this process will be used to consider next steps, which might include revisions to the content of the checklist. Data and materials from this initial trial will be publicly shared in September 2019. Data will only be provided in aggregate form and will not be parsed by individual article or by journal, so as to respect the confidentiality of responses. 

Please treat the checklist and elaboration as confidential as public release is planned for September 2019.

We would be grateful for any feedback you may have.

---

## [Decision Letter · Decision Letter 1]

10 Jul 2019

Dear Dr Spielmann,

Thank you very much for submitting your manuscript "EXP1 is critical for nutrient uptake across the parasitophorous vacuole membrane of malaria parasites" for consideration as a Research Article at PLOS Biology. Your manuscript has been evaluated by the PLOS Biology editors, an Academic Editor with relevant expertise, and by several independent reviewers.

As you will read from the reviews (below), all the reviewers agree that the study is very well done. Reviewers #2 and #3 mainly request minor changes to improve the clarity and accessibility of the work. Reviewer #1 does suggest some additional experimental analyses. We have discussed the reviewers with the Academic Editor and feel there is really only one main outstanding question that needs some addressing which Reviewer 1 asks for experimentally and Reviewer 3 poses a question about - if EXP2 has two roles, one as a nutrient channel and the other for protein export (both of which require EXP2 to be inserted into the membrane) and EXP1 is required for EXP2 to correctly localise, why then is there a nutrient uptake defect but not a protein export defect when EXP1 is knocked down? How do you propose protein export EXP2 localises to the membrane without EXP1 and what is different about the two forms of EXP2? Reviewer 1 has asked for a solubility assay to determine if there is a change in the solubility profile of EXP2 after EXP1 knockdown, which we feel is not an arduous experiment to validate that EXP1 is indeed critical for EXP2 inserting into the membrane (which is only shown by IFA) and to explain these results. Reviewer 3 has asked for an explanation for the role of EXP1 in proper localisation of EXP2. We believe the experiment may certainly help to address/resolve this question and would encourage it. Regarding the other experiments recommended by Reviewer 1 (Point 2 - pull downs with mutant forms of EXP1 to reveal which regions of this protein are important for interaction) - we acknowledge that these may be beyond the scope of this paper and will not consider it essential for publication. Additionally, Reviewer 2 asked for IFAs to analyse the localisation and fluorescent instensity of EXP1 during this time frame - once again, while welcome, we do not feel it is essential and will not insist on it for publication. Overall, we welcome resubmission of a revised manuscript that takes into account the reviewers' comments. 

Please note that we cannot make any decision about publication until we have seen the revised manuscript and your response to the reviewers' comments. Your revised manuscript is also likely to be sent for further evaluation by the reviewers.

Your revisions should address the specific points made by each reviewer. Please submit a file detailing your responses to the editorial requests and a point-by-point response to all of the reviewers' comments that indicates the changes you have made to the manuscript. In addition to a clean copy of the manuscript, please upload a 'track-changes' version of your manuscript that specifies the edits made. This should be uploaded as a "Related" file type. You should also cite any additional relevant literature that has been published since the original submission and mention any additional citations in your response. 

Before you revise your manuscript, please review the following PLOS policy and formatting requirements checklist PDF: http://journals.plos.org/plosbiology/s/file?id=9411/plos-biology-formatting-checklist.pdf. It is helpful if you format your revision according to our requirements - should your paper subsequently be accepted, this will save time at the acceptance stage.

Please note that as a condition of publication PLOS' data policy (http://journals.plos.org/plosbiology/s/data-availability) requires that you make available all data used to draw the conclusions arrived at in your manuscript. If you have not already done so, you must include any data used in your manuscript either in appropriate repositories, within the body of the manuscript, or as supporting information (N.B. this includes any numerical values that were used to generate graphs, histograms etc.). For an example see here: http://www.plosbiology.org/article/info%3Adoi%2F10.1371%2Fjournal.pbio.1001908#s5.

For manuscripts submitted on or after 1st July 2019, we require the original, uncropped and minimally adjusted images supporting all blot and gel results reported in an article's figures or Supporting Information files. We will require these files before a manuscript can be accepted so please prepare them now, if you have not already uploaded them. Please carefully read our guidelines for how to prepare and upload this data: https://journals.plos.org/plosbiology/s/figures#loc-blot-and-gel-reporting-requirements.

We expect to receive your revised manuscript within two months. Please email us (plosbiology@plos.org) to discuss this if you have any questions or concerns, or would like to request an extension. At this stage, your manuscript remains formally under active consideration at our journal; please notify us by email if you do not wish to submit a revision and instead wish to pursue publication elsewhere, so that we may end consideration of the manuscript at PLOS Biology.

When you are ready to submit a revised version of your manuscript, please go to https://www.editorialmanager.com/pbiology/ and log in as an Author. Click the link labelled 'Submissions Needing Revision' where you will find your submission record. 

Sincerely,

Hashi Wijayatilake, PhD

Managing Editor

on behalf of

Lauren A Richardson, Ph.D, 

Senior Editor

PLOS Biology

REVIEWS:

Reviewer #1: 

Review of ‘EXP1 is critical for nutrient uptake across the parasitophorous vacuole membrane of malaria parasites’ by Mesén-Ramírez et al for PLoS Biology.

Overall Comments

This is a very interesting paper which incorporates an enormous amount of work and finally gives us insight into the crucial function of EXP1, an abundant Plasmodium protein. EXP1 was thought to be involved in resisting oxidative stress but probably has a more important function in helping intraerythrocytic parasite forms acquire nutrients across their enveloping vacuole membranes. 

Major Comments

1. In Fig 1 the authors establish using a very clever selection linked integration system that loxP mediated deletion of most of the exp1 gene arrests development of parasite blood stages. Detailed phenotyping of the mutants follows indicating they progress very slowly through the cell cycle after exp1 deletion. The young ring stage EXP1 null parasites produce more PVM protrusions than normal and often ‘hug’ the RBC surface. It was not clear to me if loss of exp1 is lethal for the parasite or do they continue to grow slowly? There is no real explanation of why the ring stage ∆EXP1 parasites appear abnormal and if this appearance can by reversed by complementation with the full-length wildtype EXP1. Is the deletion of the exp1 gene incomplete and do the wildtype parasites outgrow the exp1 mutants after a few cell cycles?

2. In Fig 2 loss of EXP1* is complemented with wildtype EXP1 proteins under different strength promoters as well as with mutant forms of the EXP1 protein. This establishes that the complementing form of EXP1 needs to be strongly expressed and that residues in the transmembrane domain and ED region are important for function. While this section is informative as to what parts of EXP1 are important for parasite growth no information is provided as to why. Given that EXP1 appears critical for EXP2 nutrient pore function, pulldowns with the mutant forms of EXP1 might reveal which regions of this protein are important for interaction with EXP2. 

3. In Fig 3 the issue of whether or not EXP1 is important for protection of parasites from oxidative damage. This is because previous work reported that EXP1 was a glutathione S-transferase. The work presented here indicated that loss of EXP1 did not greatly sensitise the parasites to oxidative damage suggesting EXP1 possibly has another major function. The experiments are comprehensive and sufficiently cover the issue. 

4. In Fig 4 and S4 and it is shown the EXP1 null mutants can still export parasite proteins into the RBC compartment. Subsequent patch clamp analysis of the PVM indicates that current flow across the PVM is greatly reduced in the EXP1 null mutant. While I understand the basics of the Fig 4d what units are pA? Why does the rapalog scale show 0 pA and the control 35 pA? Was 0.2s the total measurement time or just the length of the scale bar beneath it?

5. In Fig 5 the localisation of EXP2 is examined in EXP1 null parasites and found to be aberrant being concentrated into PVM protrusions and bodies around the parasite. ETRAMP5 was similarly affected but not other ETRAMPs. The implication if this finding is that EXP2 requires EXP1 for an even localisation around the PVM and pulldown experiments indicate the two proteins interact. The result that EXP1 is required for EXP2 function is a curious finding for it would be expected to reduce both nutrient uptake as well as protein export since EXP2 forms both nutrient pores and the protein translocon pore of PTEX. Up until this EXP1 null result I would have presumed the heptameric EXP2 pore structure would be similar in both nutrient and translocon complexes but the EXP1 result suggests the nutrient pore is a distinct and possibly different structure. This could be explored by determining if EXP2 can form a membrane pore in EXP1 null parasites by determining if EXP2 is still membrane associated when EXP1 is absent (ie, does EXP2 become soluble without EXP1?). Also, does the EXP2 protein that is pulled down with EXP1 lack any association with the rest of PTEX?

6. In Fig 6 the growth of parasites was examined on nutrient limited media to show that this sensitised parasites to grow slowly particularly when EXP1 was deleted or expressed at a low level. Reduction of EXP1 expression appears to reduce the capacity of nutrients across the PVM particularly when nutrients are in limited supply. The minimal media appears to have reduced levels of amino acids but which amino acids? Only isoleucine is essential with the remaining amino acids being produced from haemoglobin breakdown. Was this experiment repeated with only isoleucine being limited? Part C should have p values between important data pairs. In part E which columns are treated with azide versus no azide and which are minimal versus complete medium. 

Minor Comments.

7. Fig 1. Part K right. What are the band sizes of the Kb ladder? Maybe label as per part C. In part L right, what do the red boxes mean? Is ‘2A’ in the gene diagram the same as T2A in the legend?

8. Indicate cell cycle number in Fig 1d and e.

9. Line 439 is ‘Light blue arrow heads show blebs.’ in part E mean to be in part F? 

10. Line 450 What gene is Lyn-Cherry?

11. In Fig 2A size bars are missing from the IFA panel.

12. What do the coloured boxes in 2D mean?

13. Line 569. I could not find Table S1.

--

Reviewer #2 (signs review as Markus Meissner): 

Summary

In this study the Spielmann group performs an extensive functional characterisation of PfEXP1. This protein has been described previously as essential for the asexual development of Plasmodium spp and multiple, potential roles, including involvement in artemisinin resistance, resistance to oxidative stress or interactions and uptake of host apolipoprotein have been suggested by others.

In this study, a conditional mutant for Exp1 was generated using the DiCre system, which allowed the authors to perform a very thorough and detailed analysis of the resulting phenotype, using a combination of very neat, cutting edge technologies. 

Briefly the authors convincingly demonstrated that:

- Exp1 is essential and parasites fail to replicate in absence of Exp1

- A carefully performed complementation analysis demonstrates that full length Exp1 is required for its essential function. Interestingly, the analysis indicates that there is only poor functional conservation within apicomplexans, since Exp1 of P.berghei appears to complement poorly for PfExp1. 

Furthermore, the previous hypothesis that EXP1 is crucial for protection against oxidative stress has been convincingly invalidated, since mutations in the GST-domain did not result in loss of complementation. Similarly, the previous suggestion that EXP1 is involved in artemisinin resistance was invalidated. 

- Loss of GST activity results only in minor effects on parasite growth and EXP1-activity must be crucial for other functions.

- Using patch Clamp measurements, the authors convincingly show that Exp1 is essential for the function of the major nutrient permeable channel of P.falciparum, a major and important finding!

- EXP1 is required for “smooth” localisation of EXP2, but not for PTEX function itself

- EXP1 and EXP2 can interact

- EXP1 is required for nutrient uptake, but not PSAC activity at the RBC membrane

Own opinion:

This is a very careful and exhaustive analysis that should be the gold-standard for a proper characterisation of phenotypes. It leaves very little room for major criticism, since the authors took great care to perform many control experiments, so that the final conclusions are fully supported by the data. In fact, the authors provide so many data and controls that it would have been sufficient for a separate paper. 

Most importantly, this study can be seen as a milestone in order to understand the dual function(s) and differential regulations of Exp1 and Exp2 in protein export and nutrient acquisition. It will also end speculations regarding additional functions/roles of Exp1 for drug resistance or resistance to oxidative stress. 

There is no doubt that it will be of high interest to a broad readership.

My comments are relatively minor, since I didn’t find any data/conclusions that require additional experimental validation.

Minor comments:

- The floxed locus is myc- and HA-tagged. Did the author also use alpha-myc to check for expression of truncated construct?

- Fig.1c: The authors present WB and PCR of cultures induced with Rap. It seems there is no significant downregulation within the first 24h and Exp1 is fully lost during the second cycle. It would be helpful if the authors could also provide IFAs to analyse the localisation and fluorescent intensity of Exp1 during this time frame.

- The description of the figures in the text is sometimes confusing. For example in line 132 its mentioned: (Fig.1f, arrows). There are 4 differently coloured arrows in Fig.1f. Other examples throughout the manuscript. This should be thoroughly corrected in the revision.

- It is recommended that the authors describe some details of their patch-clamp method presented in figure 4 already in the result session for better understanding and flow of the manuscript.

- Figure 5 is slightly overloaded. The IFAs using antibodies for EXP2 (5d,e) can easily be moved to supplementary figures, since it shows the same as 5a,b

- In the discussion the authors should also consider recent results from other apicomplexan parasites, such as Gold et al., 2015, which are nicely supporting the data presented in this study.

--

Reviewer #3: 

Review for submission of manuscript PBIOLOGY-D-19-01632R1

By Paolo Mesén-Ramírez, Bärbel Bergmann, Thuy Tuyen Tran, Matthias Garten, Jan Stäcker, Isabel Naranjo-Prado, Katharina Höhn, Joshua Zimmerberg, Tobias Spielmann

“EXP1 is critical for nutrient uptake across the parasitophorous vacuole membrane of malaria parasites”

Mesén-Ramírez et al. report their findings concerning the parasitophorous vacuolar membrane protein Exported protein-1 in the apicomplexan parasite P. falciparum causing malaria. They characterize its properties as a nutrient conducting trans-membrane channel and also its functional and physical association with Exported protein-2, another vacuolar membrane protein that functions as a protein-conducting channel in the vacuolar translocon PTEX but also serves as a nutrient-conducting transmembrane channel.

Because nutrient uptake is essential to parasite survival and its expansion inside the infected red blood cell, there is considerable interest in “channeling the vacuole” by characterizing these nutrient or protein “channels” to better understand parasite physiology. This type of studies potentially extends the number of drug-targets by identifying novel essential pathways, the so-called permeability pathways, and their molecular effectors to design much-needed new anti-malarial compounds. 

The work shows that EXP1 is needed for nutrient uptake at the vacuolar membrane, required for proper vacuolar localization of the pore-forming protein conducting channel EXP2, a core subunit of EXP2 but also a nutrient permeable channel as demonstrated by Garten et al. in a previous study.

I think this is beautiful work in terms of cellular and molecular apicomplexan parasitology that significantly expands the current knowledge of the field. I have a few questions and objections that the Authors may want to address but I strongly support publication once these issues have been addressed.

Comments and questions.

Line 37. What do authors means by “major” in major integral PVM protein EXP1? It is unclear to me. Is it abundance?

Line 41. I would suggest saying ‘pore-forming protein’ (rather than molecule)

Line 61. I would suggest being more precise “which forms a heptameric PVM-spanning channel”. It does not hurt.

Authors use conditional KO of the gene exp1 to show that the protein EXP1 is essential for propagation in RBCs. Ring stages are slow to reach the trophozoite stages and schizogony is interrupted. Thus the effects are quite drastic and remarkable. The pervasive effects of EXP1 KO even affect the gametocytes but to a lower extent. The cytology indicates the presence of what Authors call ‘blebs’ protruding out of the PVM although the integrity of the PVM is not compromised it is obviously altered at the ultra-structural level. The methodology and analysis are extremely thorough and beautifully well illustrated in the figures.

Although I find the section describing the complementation (or lack of) of the cornucopia of chimeric EXP1 constructs tested by the Authors to probe the role of the different regions in protein functionality quite complex, it is remarkable work. I am glad the Authors included Supp Fig 2 otherwise it would be a nightmare to understand it.

Paragraph starting line 167.

Based on high-resolution crystallographic structures of membrane proteins (MP), GXXG motifs are known to be essential for very close TM helix to TM helix packing/contact interactions in MP structures (whether it is intramolecularly as shown in Aquaporins for example) or between molecules to favor tightly packed oligomers inside the bilayer. Unless I missed where it is described, what kind of point mutations did the Authors introduce in the two GXXG motifs.

Line 161-163. “EXP1 lacking a 10 amino acid stretch in the N-terminus named E-domain (EXP1ΔED), a region proposed to be necessary for the dimerization based on similarity with MAPEG complemented only poorly (Fig. 2b). However, EXP1ΔED was still capable to homo-dimerize (Fig. 2e), indicating that the loss of function was not related to the capacity to oligomerize.”

Fig 2e. What is the sequence of the so-called E ‘domain’ or better said 10-aminoacid long protein segment? Why would ΔED deletions then seem to form more dimer as suggested on Fig. 2e ... if that band actually meaningful? As a membrane protein biochemist, It do not really know what to do with what the Authors show on Fig2e concerning dimerization of their constructs wtEXP1 compared to the TMmut and ΔED variants. Honestly, I am very skeptical about this. MPs display erratic electrophoretic behaviors on (denaturing) gels as a function of the detergent environment and fraction preparation (among other things). I am not convinced the Fig and results substantiate the claims made by the Authors when it comes to association state.

Based on this comment, this makes the statement on lines 172-173 a bit dubious in my opinion. Based on their extensive complementation assays I agree that all regions (rather than domains, the only domain is the TMD after all) are important to protein function(s) but I would not conclude any further.

The sections on the role of EXP1 in coping with oxidative stress or its effect on artemisin resistance are clear and the results well presented.

The GST related section is clear. I must say that as a biochemist and structural biologist that read the article in Cell (reference 16 from Lisewski et al), it makes me want to cry when I see how poorly characterized the so-called recombinant and pure EXP1 protein expressed in E. coli was in this study. One wonders if it is at all a GST…

Since the previously postulated function as a heme-detoxifying GST did not appear to be responsible for the phenotype in ΔEXP1 parasites, the Authors look at other possible pathways and possibly the interaction between EXP1 and EXP2, the PTEX subunit, and thus protein vacuolar trafficking.

If memory serves, EXP1 was not initially identified as a main interacting partner with PTEX (via EXP2 it seems based on this study) in the initial proteomic-based approaches (most from de Koning-Ward’s group). While subsequent work by this group (this work and reference 30 in 2016) shows IPs with interaction between EXP1 and EXP2. Would the Authors offer an explanation about this?

Line 249. Typo

“ΔEXP1 parasites showed NO (instead of not) defect in the export of 249 SBP1, REX1, REX2 …”

Line 307.

“Parasite surface ANION (instead of anyone) channel”

“Loss of EXP1 abolished the correct localisation of EXP2, a pore-forming molecule required for the nutrient-permeable channel activity and protein export at the PVM. Unexpectedly loss of EXP1 however affected only the nutrient-permeable channel activity of the PVM but not protein export.” Could they offer an explanation for the role of EXP1 in proper localization of EXP2?

The Authors show that EXP1 controls the localization of EXP2 but does not affect protein trafficking mediated by EXP2 through PTEX and that only the NP channel activity is defective in ΔEXP1 parasites: that activity is vacuolar not at the iRBC membrane surface. They also show that the two proteins interact in reciprocal IPs. EXP1 depleted cells are susceptible to amino-acid starvation.

All their results are clear and support their conclusions. These are delicate experiments combining powerful genetic tools in Plasmodium and patch-clamp based measurements on parasites released from their host RB cell.

They demonstrate that EXP1 is required for EXP2-based NP channel activity and functionally distinguish EXP2 roles in nutrient uptake and protein export. Does this hint again at two molecular pools of EXP2: one associated with EXP1 for nutrient uptake and one associated with the rest of the PTEX core components (PTEX150, HSP101) and other ancillary proteins?

Thus, could the Authors make a simple figure that summarizes their model on how EXP1 (and EXP2) interplay in the cell. 

They hint at their model at lines 361-362. “Thus, the expression of two functions, nutrient uptake and protein export, are at least differentially regulated and at most molecularly distinct.”

It is a model and is likely to be imperfect but at least it would summarize their findings in a simple way more easily understandable by a non-specialist reader.

---

## [Editor Report · Decision Letter 2]

22 Aug 2019

Dear Dr Spielmann,

Thank you for submitting your revised Research Article entitled "EXP1 is critical for nutrient uptake across the parasitophorous vacuole membrane of malaria parasites" for publication in PLOS Biology. 

The Academic Editor and I have now assessed your revision and find it exceptionally well-done. We're delighted to let you know that we're now editorially satisfied with your manuscript. We will publish your study, assuming you are willing to make the final edits to meet our production requirements. Congratulations!

However before we can formally accept your paper and consider it "in press", we also need to ensure that your article conforms to our guidelines. A member of our team will be in touch shortly with a set of requests. As we can't proceed until these requirements are met, your swift response will help prevent delays to publication.

Please note that you may have the opportunity to make the peer review history publicly available. The record will include editor decision letters (with reviews) and your responses to reviewer comments. If eligible, we will contact you to opt in or out.

Sincerely,

Lauren A Richardson, Ph.D

Senior Editor

PLOS Biology

DATA POLICY:

**Please also ensure that figure legends in your manuscript include information on where the underlying data can be found.**

For manuscripts submitted on or after 1st July 2019, we require the original, uncropped and minimally adjusted images supporting all blot and gel results reported in an article's figures or Supporting Information files. We will require these files before a manuscript can be accepted so please prepare them now, if you have not already uploaded them. Please carefully read our guidelines for how to prepare and upload this data: https://journals.plos.org/plosbiology/s/figures#loc-blot-and-gel-reporting-requirements.

---

## [Editor Report · Decision Letter 3]

10 Sep 2019

Dear Dr Spielmann,

On behalf of my colleagues and the Academic Editor, Tania F de Koning-Ward, I am pleased to inform you that we will be delighted to publish your Research Article in PLOS Biology. 

Early Version

PRESS 

Kind regards,

Sofia Vickers

Senior Publications Assistant

PLOS Biology

On behalf of, 

Lauren Richardson,

Senior Editor

PLOS Biology